# QuantV2X: A Fully Quantized Multi-Agent System for Cooperative Perception

## Abstract

Cooperative perception through Vehicle-to-Everything (V2X) communication offers significant potential for enhancing vehicle perception by mitigating occlusions and expanding the field of view. However, past research has predominantly focused on improving accuracy metrics without addressing the crucial system-level considerations of efficiency, latency, and real-world deployability. Noticeably, most existing systems rely on full-precision models, which incur high computational and transmission costs, making them impractical for real-time operation in resource-constrained environments. In this paper, we introduce **QuantV2X**, the first fully quantized multi-agent system designed specifically for efficient and scalable deployment of multi-modal, multi-agent V2X cooperative perception. QuantV2X introduces a unified end-to-end quantization strategy across both neural network models and transmitted message representations that simultaneously reduces computational load and transmission bandwidth. Remarkably, despite operating under low-bit constraints, QuantV2X achieves accuracy comparable to full-precision systems. More importantly, when evaluated under deployment-oriented metrics, QuantV2X reduces system-level latency by 3.2× and achieves a +9.5 improvement in mAP30 over full-precision baselines. Furthermore, QuantV2X scales more effectively, enabling larger and more capable models to fit within strict memory budgets. These results highlight the viability of a fully quantized multi-agent intermediate fusion system for real-world deployment. The system will be publicly released to promote research in this field. Please refer to the supplementary materials for the demo webpage and codebase.

## 1 Introduction

Vehicle-to-Everything (V2X) cooperative perception has emerged as a promising paradigm for enabling safe and intelligent autonomous driving (Li et al., 2021b; Zhou et al., 2025; Lei et al., 2025b). By allowing autonomous agents to share real-time sensor information, it creates a collective perception system that extends beyond the field of view of any single vehicle, significantly enhancing situational awareness for all agents (Zhao et al., 2024; Zhou et al., 2024b). Despite remarkable progress in model design and accuracy improvements, most prior work has been developed under full-precision (FP32) assumptions, leading to prohibitive computational, memory, and communication costs. As illustrated in Fig. 1, full-precision systems cannot be accommodated within the tight memory budgets of in-vehicle GPUs without aggressive model size reduction, which inevitably sacrifices the model's expressiveness and causes substantial performance degradation. Moreover, even when such models are deployed, the high inference latency of full-precision networks, compounded by the transmission overhead of sharing FP32 BEV features, introduces prohibitive system-level delays. These intertwined bottlenecks highlight a critical gap between algorithmic advances in cooperative perception and their practical feasibility for real-world autonomous driving deployments.

To bridge this gap, we present **QuantV2X**, a fully quantized multi-agent cooperative system designed to holistically address the system-level latency and performance drop in resource-constrained V2X settings (shown in Fig. 1). Our core insight is that full-precision representations in both local computation and agent-to-agent communication dominate end-to-end latency and lead to downstream performance drop. Building on this, QuantV2X delivers a full-stack recipe encompassing both model-side and communication-side efficiency. On the model side, we propose a post-training quantization (PTQ) process that transforms pretrained full-precision models into compact low-bit

Figure 1: Motivation. *Left:* Full-precision cooperative perception systems are ill-suited for real-world deployment. *Right:* QuantV2X presents an efficient and scalable solution for real-world cooperative driving systems.

networks while maintaining competitive accuracy. To mitigate the challenges brought by quantization-induced feature degradation, we introduce a novel alignment module that jointly corrects spatial misalignment and feature distribution shifts among heterogeneous agents. On the communication side, we replace costly FP32 BEV feature transmission with compact low-bit messages, where each agent transmits only the code indices of a shared codebook. These indices act as quantized representations of the full feature map, allowing the receiver to reconstruct high-fidelity features locally while significantly reducing communication bandwidth during collaborations. Together, under real-world latency constraints, QuantV2X reduces end-to-end system latency by $\times$**3.2** compared to the full-precision system with **9.5%** mAP30 performance improvements in V2X-Real dataset. These results demonstrate that QuantV2X effectively overcomes system-level efficiency bottlenecks and enables real-time, high-performance V2X cooperative perception.

The experimental sections move beyond the conventional accuracy-centric paradigm, focusing on evaluating the holistic performance of the system in realistic deployment scenarios. In Section 3.1, we show that QuantV2X maintains the perception ability of full-precision systems, preserving up to 99.8% of their accuracy even under INT4 weight and INT8 activation quantization. In Section 3.2, we further show that QuantV2X consistently surpasses full-precision systems when evaluated under system-level latency, highlighting its real-world efficiency. Finally, in Section 3.3, we illustrate how quantized deployment enables larger and more capable models to run on edge platforms without exceeding resource budgets, thereby expanding both system capacity and performance. Collectively, these results position QuantV2X as a practical and scalable pathway towards fully deployable multi-agent systems for V2X cooperative perception.

**Contribution.** In this work, we address the problems of inefficiency and performance degradation for cooperative perception in real-world resource-constrained scenarios. We illustrate the system-level latency bottleneck in full-precision systems and introduce **QuantV2X**, *a fully quantized multi-agent system for cooperative perception* that enables efficient model inference and multi-agent communication with maximum perception performance preservation while meeting the requirements of real-world deployment. To the best of our knowledge, this is the first work to demonstrate the viability and practicality of a fully quantized intermediate fusion system for future real-world deployment.

## 2 METHODOLOGY

### 2.1 QUANTV2X: A SYSTEM OVERVIEW

As shown in Fig. 2, QuantV2X presents a fully quantized system that unifies efficiency at both the model level and the communication level. QuantV2X consists of three stages: (i) **a full-precision pretraining stage**, where we train a full-precision (FP32) cooperative perception model that serves as the foundation for subsequent quantization, (ii) **a codebook learning stage**, where the model learns quantized transmission feature representations for communication-efficient collaboration, and

Figure 2: QuantV2X overview. On the model side, QuantV2X transforms full-precision neural networks into compact low-bit representations, reducing computational overhead without sacrificing accuracy. On the communication side, it replaces bandwidth-heavy floating-point feature maps with quantized message representations, enabling efficient collaborations under strict transmission budgets.

(iii) **a post-training quantization (PTQ) stage**, where both full-precision models and features are converted into low-bit formats with minimal accuracy degradation.

## 2.2 FULL-PRECISION PRETRAINING

In this stage, we adopt an intermediate fusion architecture. Let $b$ represent an agent type. For the $i$th agent in the set $S_{[b]}$, we define $f_{\text{encoder}[b]}$ as its perception encoder, input $\mathbf{O}_i$ as its raw input (RGB images or LiDAR point clouds) and $\mathbf{B}_i$ as its final detection output. The operation of the $i$th agent works as follows:

$$\mathbf{F}_i = f_{\text{encoder}[b]}\left(\mathbf{O}_i\right), \quad \mathbf{F}_{j\to i} = \Gamma_{j\to i}\left(\mathbf{F}_j\right), \quad \mathbf{H}_i = f_{\text{fusion}}\left(\left\{\mathbf{F}_{j\to i}\right\}_{j\in S_{[b]}}\right), \quad \mathbf{B}_i = f_{\text{head}}\left(\mathbf{H}_i\right),$$

where $\mathbf{F}_i$ is the initial BEV feature map produced by the encoder, $\Gamma_{j\to i}(\cdot)$ is an operator that transmits $j$th agent's feature to the $i$th agent and performs spatial transformation, $\mathbf{F}_{j\to i}$ is the spatially aligned BEV feature in $i$th's coordinate frame (note that $\mathbf{F}_{i\to i} = \mathbf{F}_i$), $\mathbf{H}_i$ is the fused feature and $\mathbf{B}_i$ is the final detection output obtained by a detection head $f_{\text{head}}(\cdot)$. This stage learns a complete FP32 model, parameterized by $f_{\text{encoder}[b]}, f_{\text{fusion}}, f_{\text{head}}$.

## 2.3 CODEBOOK LEARNING

### 2.3.1 QUANTIZED MESSAGE REPRESENTATION VIA CODEBOOK

The transmission of FP32 BEV features poses major challenges for cooperative perception, incurring high bandwidth and computation costs on resource-limited hardware, as evidenced by recent deployments (Xiang et al., 2025). Motivated by the approaches proposed in (Han et al., 2015; Hu et al., 2024), we employ a codebook-based messaging approach for collaborative agents and introduce a novel quantization-aware codebook learning method optimized for fully quantized systems. The codebook can be seen as a dictionary data structure represented by {codebook index : codebook feature}. Formally, we define the codebook as a learnable dictionary $\mathcal{D} = \{d_1, d_2, \ldots, d_{n_L}\} \in \mathbb{R}^{C\times n_L}$, where each entry $d_\ell \in \mathbb{R}^C$ represents a $C$-dimensional feature vector and $n_L$ represents the number of codes. The codebook is shared across all agents and serves as a compressed basis for BEV features. During collaboration, each agent transmits only the codebook indices to other agents instead of transmitting BEV feature maps. Regarding communication volume, for a BEV feature of dimensions $H \times W \times C$, the original communication bandwidth requirement can be computed as $\log_2\left(H \times W \times C \times 32/8\right)$, where the number 32 indicates FP32 data representation, and the division by 8 converts bits to bytes. In contrast, when employing the codebook index representation with a codebook, the bandwidth is reduced as $\log_2\left(H \times W \times \log_2(n_L) \times n_R/8\right)$, where $\log_2(n_L)$ denotes the number of bits required to represent each code index integer, and $n_R$ indicates the number of codes used.

During transmission, given a BEV feature vector $F_{[h,w]} \in \mathbb{R}^C$ at spatial location $(h, w)$, the nearest code in the dictionary is selected via:

$$\text{index}_{[h,w]} = \arg\min_{\ell\in\{1,2,\ldots,n_L\}} \left\|F_{[h,w]} - d_\ell\right\|_2^2. \tag{1}$$

When using multiple codes per location ($n_R > 1$), the reconstructed feature vector $\hat{F}_{[h,w]}$ is computed as a weighted combination of selected codes:

$$\hat{F}_{[h,w]} = \sum_{r=1}^{n_R} \alpha_r \cdot d_{\text{index}_r},\tag{2}$$

where $\alpha_r$ are the combination weights and $\{\text{index}_r\}_{r=1}^{n_R}$ are the corresponding selected code indices. The combination weight is generated by computing the distances between input feature segments and all codebook entries, converting these distances to logits, and applying Gumbel-Softmax to produce soft, differentiable weights during training (which become hard one-hot selections during inference), which are then used to reconstruct the quantized feature as a weighted combination of the selected codes from each codebook group.

### 2.3.2 CODEBOOK TRAINING STRATEGY

We train the codebook in two stages. In the first stage, we randomly initialize $\mathcal{D}$ and freeze all other model parameters. Given frozen BEV features $F \in \mathbb{R}^{H \times W \times C}$ extracted from the encoder pretrained from full-precision models in Section 2.2, we assign each spatial position $(h, w)$ to one or more code indices via a nearest-neighbor assignment (argmin of squared Euclidean distance) within a product-quantized codebook. The learning objective becomes the following.

$$\min_{\Theta_{\text{cb}}} \sum_{(h,w)} \left\| F_{[h,w]} - \hat{F}_{[h,w]} \right\|_2^2,\tag{3}$$

where $\Theta_{\text{cb}}$ denotes all the parameters within the codebook module.

In the second stage, we unfreeze all model parameters and jointly optimize the encoder, fusion module, detection head, and codebook. The encoder is now trained to produce BEV features $F$ that are naturally quantizable with $\mathcal{D}$. At each forward pass, the BEV features are quantized to $\hat{F}$ using the current codebook, and the detection is performed on the quantized representation. The joint objective becomes:

$$\min_{\theta, \mathcal{D}} \mathcal{L}_{\text{det}}(\hat{B}, B^{\text{gt}}) + \lambda_{\text{rec}} \sum_{(h,w)} \left\| F_{[h,w]} - \hat{F}_{[h,w]} \right\|_2^2,\tag{4}$$

where $\theta$ denotes all the model parameters excluding $\mathcal{D}$, $\mathcal{L}_{\text{det}}$ denotes the standard detection loss, $\hat{B}$ denotes the detection output computed from quantized features $\hat{F}$, $B^{\text{gt}}$ denotes the ground-truth bounding boxes, and $\lambda_{\text{rec}}$ controls the weight of the reconstruction term.

## 2.4 POST-TRAINING QUANTIZATION

The goal of the post-training quantization stage is to convert the full-precision model into a low-bit format while minimizing performance degradation. This stage only requires a small fraction of calibration data and does not need to retrain the whole model. We quantize both individual tensors and full network modules, leveraging deployment-friendly techniques compatible with inference engines like TensorRT (Migacz, 2017). Unlike prior work (Zhou et al., 2024a), which only partially quantizes the network, we apply end-to-end quantization across the entire pipeline (from the encoder and fusion module to the detection decoder) to achieve a fully quantized system.

### 2.4.1 PRELIMINARIES: QUANTIZATION FOR TENSORS

Quantization maps floating-point (FP) values $x$ (e.g., weights or activations) to low-precision integer approximations $x_{\text{int}}$ following:

$$x_{\text{int}} = \text{clamp}\left( \left\lfloor \frac{x}{s} \right\rceil + z, q_{\min}, q_{\max} \right),\tag{5}$$

where $\lfloor \cdot \rceil$ denotes rounding-to-nearest integer, introducing rounding error $\Delta_r$; $z$ denotes the zero-point and $s$ denotes the scale factor defined as:

$$s = \frac{q_{\max} - q_{\min}}{2^b - 1},\tag{6}$$

where $b$ is the target bit-width. The clamp operation ensures the result lies within the quantization range $[q_{\min}, q_{\max}]$, introducing a clipping error $\Delta_c$. The dequantized approximation of the original FP values is obtained via:

$$\hat{x} = s \cdot (x_{\text{int}} - z). \tag{7}$$

### 2.4.2 QUANTV2X CALIBRATION PROCEDURE

Calibration is essential in our fully quantized system to ensure that the transition from full-precision to quantized models does not hurt performance. The calibration procedure is outlined in Algorithm 1. We first construct a sampled subset from the training dataset as a calibration dataset. To address the variability in cooperative interactions, we introduce a multi-agent sampling strategy that randomly samples agent combinations and communication patterns for the construction of the calibration dataset. By exposing the model to varying numbers and configurations of interacting agents, we ensure that the quantization parameters are robustly calibrated to reflect the dynamic and heterogeneous nature of real-world cooperative perception.

---

**Algorithm 1** QuantV2X Calibration

---

**Input**: Pretrained FP model with $N$ blocks, Calibration dataset $D^c$, Iteration $T$. Blocks denote the perception network components (e.g., backbone, fusion module, downstream head).

**Output**: Quantization parameters of both activation and weight in the network: weight scale $s_w$, weight zero-point $z_w$, activation scale $s_a$, and activation zero-point $z_a$.

1: **for** $B_n = \{B_i | i = 1, 2, ...N\}$ **do**
2:     Initialize weight parameters $s_w$ and $z_w$ of each layer in $B_n$ using Eq.( 6);
3: **end for**
4: Use weight quantization parameters to formulate a mirrored Quantized model with N blocks;
5: Input $D^c$ to FP model to collect final output prediction $O_{fp}$;
6: **for** $B_n, B_n^q = \{B_i, B_i^q | i = 1, 2, ...N\}$ where $B_n^q$ belongs to quantized model **do**
7:     Input $D^c$ into both FP and Quantized models and collect block input $I^q$ from $B_i^q$ and block output $A_i$ from $B_i$;
8:     Input $I^q$ into $B_i^q$ to initialize activation parameters $s_a$ and $z_a$ using Eq. ( 6);
9:     **for** all $j = 1, 2, ..., T$ iteration **do**
10:         Input $I^q$ to $B_i^q$ to get block output $\hat{A}_i$;
11:         Optimize parameters $s_w, z_w, s_a$, and $z_a$ of block $B_i^q$ using Eq.( 8);
12:         **if** $B_i$ belongs to the fusion module **then**
13:             Input $\hat{A}_i$ to the following FP network to get output $\hat{O}_{par}$ of partial-quantized network;
14:             Check $\hat{A}_i$ and $A_i$ to calculate $\mathcal{L}_{\text{hetero}}$ to perform heterogeneity alignment using Eq.( 9);
15:             Check $\hat{O}_{par}$ and $O_{fp}$ to calculate $\mathcal{L}_{\text{spatial}}$ to perform spatial alignment using Eq.( 10);
16:             Optimize parameters $s_w, z_w, s_a$, and $z_a$ of layer $B_i^q$ to minimize $\mathcal{L}_{\text{hetero}}$ and $\mathcal{L}_{\text{spatial}}$;
17:         **end if**
18:     **end for**
19: **end for**

---

As calibration begins, we initialize both weight and activation quantization using the Max-min calibration strategy, which defines the quantization range based on the observed minimum and maximum values of the input tensor. This strategy aims to preserve the fine-grained structure of sparse point cloud features while remaining effective for RGB features (Zhou et al., 2024a). Given an input tensor $X$, the quantization range is set to $[X_{\max}, X_{\min}]$ and the initial quantization scale $s_0$ is then computed using Eq. 6. To enable fine-grained calibration, we linearly discretize a range around the initial scale factor $s_0$, forming a set of candidate quantization scales $\{s_t\}_{t=1}^{T}$ within the interval $[\alpha s_0, \beta s_0]$. The hyperparameters $\alpha$, $\beta$, and $T$ control the search span and resolution. We then select the optimal quantization scale $s_{\text{opt}}$ by minimizing the reconstruction error between the original and quantized representations:

$$s_{\text{opt}} = \arg\min_{s_t} \|X - \hat{X}(s_t)\|_F^2, \tag{8}$$

where $\|\cdot\|_F$ denotes the Frobenius norm and $\hat{X}(s_t)$ represents the quantized tensor under scale $s_t$. To further reduce the rounding error $\Delta_r$, we adopt a learnable rounding strategy inspired by AdaRound (Nagel et al., 2020), introducing an auxiliary variable for each weight element to adaptively select rounding directions.

During calibration, we propose a block-wise reconstruction strategy that minimizes block-level discrepancies between full-precision and quantized outputs, reducing the interface errors that arise

with per-layer calibration. This strategy is applied across the entire multi-agent system, including $f_{\text{encoder}[b]}, f_{\text{fusion}}, f_{\text{head}}$, to keep the quantized system aligned with the full-precision reference. For multi-agent fusion $f_{\text{fusion}}$, we add an alignment module within the intermediate fusion layers to preserve cross-agent feature consistency during fusion.

### 2.4.3 ALIGNMENT MODULE

The fusion module serves as the central component of cooperative perception models, where features from all agents are aggregated into a unified representation. However, this process is particularly vulnerable to quantization noise. Directly applying conventional quantization techniques at this stage often leads to compounded feature misalignment that distorts the fused representation. For example, as illustrated in Fig. 3, naive linear quantization introduces a significant distribution shift relative to the full-precision model, ultimately harming downstream perception performance. To mitigate this quantization-induced degradation, we propose an alignment module that addresses two key sources of misalignment in cooperative perception scenarios: (i) sensor modality and architecture heterogeneity - differences in sensors (i.e., RGB and LiDAR point cloud) and encoder backbone (i.e., PointPillar (Lang et al., 2019) and SECOND (Yan et al., 2018)), and (ii) spatial discrepancies arising from real-world deployment issues such as transmission latency and pose noise due to temporal asynchrony. The alignment module mainly applies at the fusion stage with the following formulations:

**Heterogeneity Alignment Loss.** Heterogeneity among agents introduces ambiguity in activation range scaling during the calibration process. To encourage consistency between full-precision and quantized fused feature maps, we introduce a heterogeneity alignment loss based on the Kullback-Leibler (KL) divergence:

$$\mathcal{L}_{\text{hetero}} = D_{\text{KL}}\left(\mathbf{H}_i^{\text{fp}} \,\|\, \mathbf{H}_i^{\text{int}}\right), \tag{9}$$

where $\mathbf{H}_i^{\text{fp}}$ and $\mathbf{H}_i^{\text{int}}$ denote the respective fused BEV features from the full-precision and quantized models.

**Spatial Alignment Loss.** Precision loss introduced by the quantization process increases the sensitivity of the final detection output to real-world noises. To reduce the discrepancy in detection outputs due to spatial misalignment, we define a spatial alignment loss using L2 loss over the predicted bounding box distributions:

$$\mathcal{L}_{\text{spatial}} = \left\|\mathcal{B}_i^{\text{fp}} - \mathcal{B}_i^{\text{int}}\right\|_2^2, \tag{10}$$

where $\mathcal{B}_i^{\text{fp}}$ and $\mathcal{B}_i^{\text{int}}$ denote the respective bounding box representations from the full-precision and quantized models.

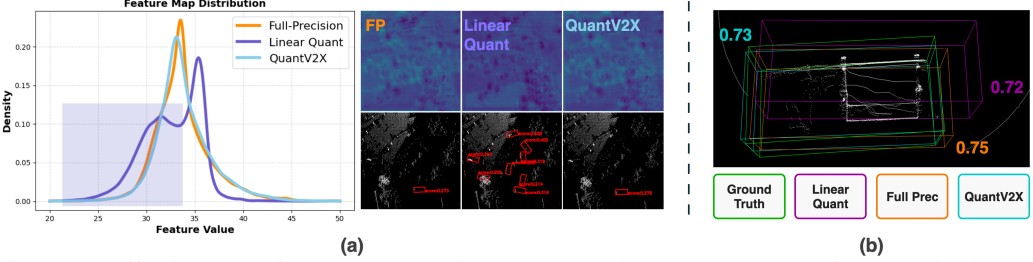

(a) (b)

Figure 3: Effectiveness of the proposed alignment module. Compared to naive quantization, (a) $\mathcal{L}_{\text{hetero}}$ leads to fewer false positive detections, and (b) $\mathcal{L}_{\text{spatial}}$ enables the quantized model to output 3D bounding boxes with more precise coordinates and higher confidence score.

## 3 EXPERIMENTS

We evaluate QuantV2X on a suite of tasks to answer the following research questions: 1) Can QuantV2X preserve perception accuracy under aggressive low-bit quantization? 2) Does QuantV2X effectively reduce real-world system-level latency and improve overall performance? 3) Can QuantV2X enable larger and more capable models under constrained GPU memory budgets?

## 3.1 MODEL-LEVEL EXPERIMENTS

**Experiment Protocols.** The main goal of the model-level experiments is to evaluate the performance of our PTQ process described in Section 2.4. To assess the effectiveness of our method in recovering spatial features, we exclude the compressor module in the cooperative perception model, which will be analyzed separately in the system-level experiments presented in Section 3.2.

**Datasets.** QuantV2X is evaluated with two real-world datasets, namely DAIR-V2X (Yu et al., 2022) and V2X-Real (Xiang et al., 2024), and one simulation dataset OPV2V (Xu et al., 2022c). DAIR-V2X exhibits one vehicle and one infrastructure, both equipped with a LiDAR with different channel numbers and a $1920 \times 1080$ camera. V2X-Real is a large-scale, real-world V2X dataset that encompasses all V2X collaboration modes with two vehicles and two roadside units. Following previous protocols (Lu et al., 2024; Xiang et al., 2025; 2024), the evaluation metric is presented as Average Precision (AP) with different intersection-over-union (IoU) thresholds. Additional evaluations are presented in the supplementary materials.

**Implementation Details.** We follow (Lu et al., 2024) and define the following notations for different agent modalities. $L_P$ denotes an agent with LiDAR sensor using the PointPillar (Lang et al., 2019) backbone, and $L_S$ denotes an agent with LiDAR sensor using the SECOND (Yan et al., 2018) backbone. $C_R$ denotes a camera-based agent with Lift-Splat-Shoot (Philion and Fidler, 2020) projection deployed and a ResNet50 model as the image encoder. Pyramid Fusion (Lu et al., 2024) is the main intermediate fusion method for our experiments, as it has the best perception performance and fastest inference time. All experiments are calibrated using 0.5% of the original training data. Additional experiments on calibration settings are presented in the supplementary materials.

### 3.1.1 GENERALIZABILITY ACROSS DIFFERENT FUSION METHODS

Table 1 shows our PTQ method generalizes well across various fusion methods, including computation-based fusion methods (Chen et al., 2019a; Xu et al., 2022c), CNN-based fusion methods (Lu et al., 2024; Liu et al., 2020b), and attention-based fusion methods (Xu et al., 2022b; Hu et al., 2022). Detailed analysis of the quantization effect on each fusion method will be provided in the supplementary materials.

Table 1: Generalizability of QuantV2X across different fusion methods. Results displayed in terms of AP30/50 on DAIR-V2X dataset (collaboration mode: $L_P + C_R$).

| Bits (W/A) | Pyramid Fusion | F-Cooper | AttFuse | V2X-ViT | Who2com | Where2comm |
|---|---|---|---|---|---|---|
| 32/32 | 75.1/68.2 | 64.5/56.0 | 68.8/63.1 | 57.4/49.5 | 63.2/57.3 | 62.1/53.7 |
| 8/8 | 74.6/67.8 | 62.9/55.4 | 67.0/61.9 | 40.0/11.0 | 59.1/54.2 | 59.5/51.8 |
| 4/8 | 74.2/66.7 | 57.4/49.5 | 66.6/60.8 | 29.9/8.8 | 57.2/52.8 | 60.4/51.5 |

### 3.1.2 COMPONENT ANALYSIS

We begin by analyzing the individual components of QuantV2X to quantify their contributions in Table 2. It can be observed that the alignment module boosts the performance recovery from 97.4% to 98.8% and 95.2% to 99.8% for $L_P + C_R$ and $L_P + L_S$ settings in terms of AP30, respectively. This component-wise evaluation leads to two key observations:

**(i) QuantV2X preserves perception capability under heterogeneous settings.** Applying a basic channel-wise linear quantization method (as described in Eq. 5) leads to a significant drop in precision and results in blurred BEV feature boundaries, as shown in Fig. 3 (a). In contrast, our heterogeneity alignment loss aligns the activation range of BEV features from heterogeneous inputs, producing sharper BEV feature maps and reducing false positives.

Table 2: Component Analysis of QuantV2X in DAIR-V2X dataset. Bits (W/A) is set to INT4/8.

| Method | AP30/50 | |
|---|---|---|
| | $L_P + C_R$ | $L_P + L_S$ |
| Full-Precision | 75.1/68.2 | 80.3/76.1 |
| Max-min | 73.2/61.5 | 76.5/60.1 |
| +AdaRound (Nagel et al., 2020) | 72.8/65.1 | 80.1/74.2 |
| +$\mathcal{L}_{hetero}$ | 74.0/66.4 | 80.8/75.3 |
| +$\mathcal{L}_{spatial}$ | 74.2/66.7 | 80.2/75.5 |

**(ii) QuantV2X demonstrates strong robustness under noisy environments.** As illustrated in Fig. 4, we evaluated the robustness of our method against localization error in the DAIR-V2X dataset. We follow the standard evaluation protocols Xu et al. (2022c); Zhou et al. (2024b); Lu et al. (2024) and use sample noises from Gaussian distribution added to the ground truth pose of each collaborating agent (positional or heading error). Under extreme settings, QuantV2X maintains performance comparable to full-precision models. Notably, it also preserves

far-range detection capability. This finding highlights the importance of incorporating a spatial alignment loss during the calibration process. Without this design, the vanilla linear quantization method fails significantly under noisy conditions. Fig. 3 (b) visualizes that the spatial alignment loss further refines the 3D bounding box predictions by correcting their coordinates.

**Ablation Study: Comparison with other PTQ methods.** The performance and computation efficiency comparisons are conducted with other PTQ methods, namely PD-Quant (Liu et al., 2022) and LiDAR-PTQ (Zhou et al., 2024a) on the DAIR-V2X dataset. As shown in Table 3, our methods achieve less performance gap compared to the full-precision model while requiring much less calibration time compared to (Zhou et al., 2024a).

Table 3: Performance comparison of PTQ methods in DAIR-V2X dataset.

| Method | Bits (W/A) | AP30 | AP50 | GPU/hr |
|---|---|---|---|---|
| Full Precision | 32/32 | 75.1 | 68.2 | – |
| PD-Quant (Liu et al., 2022) | 4/8 | 65.5 | 56.1 | 0.37 |
| LiDAR-PTQ (Zhou et al., 2024a) | 4/8 | 73.8 | 65.7 | 0.93 |
| QuantV2X (Ours) | 4/8 | 74.2 | 66.7 | 0.38 |

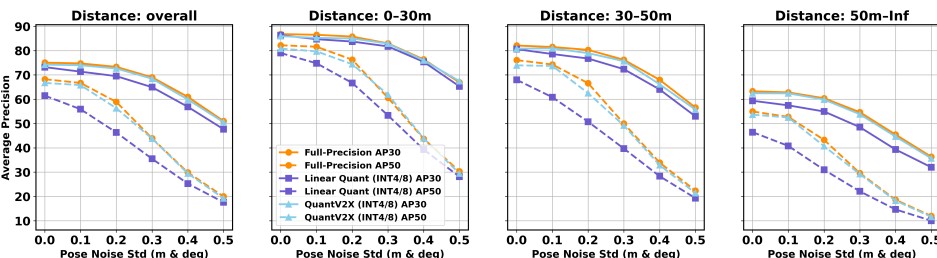

Figure 4: Robustness under pose errors (Collaboration mode: $\mathbf{L_P + C_R}$, Bits (W/A) are set to INT4/8). Note that the vanilla quantization method suffers from considerable precision loss when the pose error is enlarged.

## 3.2 SYSTEM-LEVEL EXPERIMENTS

**Experiment Protocols.** The goal of system-level experiments is to examine the performance of the quantized system considering system-level latency, including local inference latency, multi-agent communication latency, and fusion inference latency. This differs from the model-level experiments in Section 3.1, which assume the multi-agent system is ideal and well-synchronized. In the system-level setting, the cooperative perception models incorporate a compressor module for transmission. For QuantV2X, we employ the quantized message representation described in Section 2, while full-precision baselines transmit compressed BEV features unless otherwise noted. Detailed testing settings and comparisons of power consumption are reported in the supplementary materials.

**Implementation Details.** Pyramid Fusion (Lu et al., 2024) is our main fusion method as it has the best perception performance and the fastest inference time. We only consider PointPillar (Lang et al., 2019) models as each agent's backbone to be consistent with benchmarking in V2X-Real and V2X-ReaLO. The evaluations of the quantized system are conducted in terms of INT8 weight and INT8 activation to be consistent with the previous protocol (Zhou et al., 2024a).

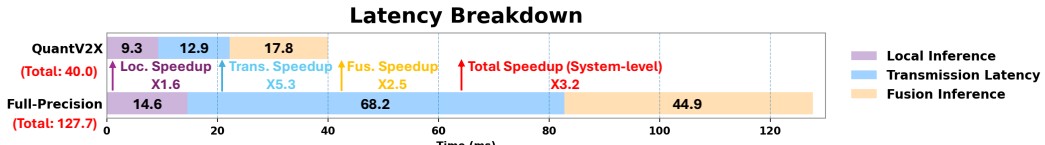

Figure 5: System-level latency breakdown (unit: ms). Note that the numbers are obtained through averaging multiple runs in real-world deployment environment.

### 3.2.1 SYSTEM-LEVEL LATENCY MEASUREMENT

To show the inference efficiency improvements of QuantV2X under real-world V2X testing environments, we evaluate the system-level latency of QuantV2X using the ROS and TensorRT platform (Xiang et al., 2025; Migacz, 2017) and compare it against a full-precision baseline. The end-to-end system-level latency ($T_{\text{sys}}$) of a cooperative perception system can be decomposed into three primary components: (i) local inference latency ($T_{\text{local}}$), representing the time each agent takes to process its own sensor data; (ii) communication latency ($T_{\text{comm}}$), the time required to transmit information

Table 4: System-level performance comparisons among different systems in V2X-Real Dataset. $\Delta$ denotes the difference with the Upper-bound, which assumes an ideal setting without considering system-level latency and transmission feature compression.

| System | Transmission Feature/Size | mAP30/50 | $\Delta$ |
|---|---|---|---|
| Upper-bound | - | 53.8/43.5 | - |
| Full-Precision (Lu et al., 2024) | BEV Feature/8.6 MB (No Compression) | 43.1/34.8 | -10.7/-8.7 |
| | BEV Feature/0.54 MB ($\times$16 Compression) | 48.8/38.0 | -5.0/-5.5 |
| Where2Comm (Hu et al., 2022) | BEV Feature/0.30 MB ($\times$16 Compression) | 49.7/39.0 | -4.1/-4.5 |
| CodeFilling (Hu et al., 2024) | Codebook/0.03 MB | 51.4/40.8 | -2.4/-2.7 |
| QuantV2X (Ours) | Codebook/0.03 MB | **52.6/42.2** | -1.2/-1.3 |

between agents; and (iii) fusion inference latency ($T_{\text{fus}}$), the time taken to process received data and generate a final perception output. A detailed testing environment will be provided in the supplementary materials. As illustrated in Fig. 5, quantization significantly reduces latency across all components: $T_{\text{local}}$, $T_{\text{comm}}$, and $T_{\text{fus}}$. These gains stem from low-precision computation for model inference and reduced communication payload between agents. In the following sections, the impact of these improvements on system-level performance is further analyzed.

### 3.2.2 SYSTEM-LEVEL PERFORMANCE EVALUATIONS

**Evaluation Setting.** To simulate the impact of system-level latency under realistic conditions, we follow the protocols in (Xu et al., 2025; Rauch et al., 2011; Xu et al., 2022b; Arena and Pau, 2019). The total system latency is defined as $T_{\text{sys}} = T_{\text{local}} + T_{\text{comm}} + T_{\text{fus}}$, where $T_{\text{local}}$ and $T_{\text{fus}}$ are obtained from Fig. 5 and $T_{\text{comm}}$ is calculated according to the transmission delay formula established by previous protocols (Xu et al., 2025; Rauch et al., 2011; Xu et al., 2022b; Arena and Pau, 2019). The communication latency is calculated as $T_{\text{comm}} = f_s/v + \mathcal{U}(0, 200)$, whereas $f_s$ is the feature size and $v$ denotes the transmission rate (which is set to 27 Mbps according to (Arena and Pau, 2019; Xu et al., 2022b)) and $\mathcal{U}$ denotes the system-wise asynchronous delay following a uniform distribution between 0 and 200 ms. All experiments are conducted on the V2X-Real dataset to remain consistent with the measurements in Section 3.2.1.

**System-level experimental results.** We compare the system-level performance of full-precision systems, Where2Comm (Hu et al., 2022), CodeFilling (Hu et al., 2024), and QuantV2X. Notably, both full-precision systems and Where2Comm are affected by latency at both the model and communication levels, whereas CodeFilling is more heavily impacted by model-level inefficiency. Table 4 presents that our method consistently outperforms the full-precision system due to the significant system-level latency reduction. Furthermore, the comparison with CodeFilling (Hu et al., 2024) emphasizes the critical role of reducing inference latency to the whole multi-agent system. These results demonstrate that in dynamic scenarios, the *information timeliness* advantage brought by low latency is sufficient to compensate for and even surpass the minor accuracy loss introduced by quantization, underscoring the importance of system-level optimization.

**Decomposition analysis of system-level latency and performance.** To rigorously disentangle the impact of model-level and communication-level inefficiencies on system performance, we conduct a component-wise decomposition analysis as shown in Fig. 6. We decouple the total system-level latency into computation latency ($T_{\text{comp}} = T_{\text{local}} + T_{\text{fus}}$) and communication latency ($T_{\text{comm}}$) and study their corresponding impacts on final system-level performance:

**(i) Impact of model-level efficiency ($T_{\text{comp}}$).** As illustrated in Fig. 6(a), model quantization addresses the computational bottleneck. For Pyramid Fusion, this reduces computation overhead ($T_{\text{comp}}$) by approximately 55% (59.5ms $\rightarrow$ 27.1ms). Crucially, this step yields a secondary benefit: the bit-width reduction (FP32 $\rightarrow$ INT8) simultaneously lowers transmission latency ($T_{\text{comm}}$) from 286.8ms to 87.0ms. This aggregate latency reduction drives immediate accuracy gains (43.1 $\rightarrow$ 48.7 mAP30), validating that efficient low-precision computation enhances overall system performance.

**(ii) Impact of communication-level efficiency ($T_{\text{comm}}$).** Complementing quantization, our communication-level optimization further reduces $T_{\text{comm}}$ from 286.8ms to 12.9ms (Pyramid Fusion). When integrated with model quantization, the total system latency drops significantly from 346.3ms to 40.0ms. This efficiency enables QuantV2X to achieve 52.6 mAP30, approaching the upper-bound of 53.8 and demonstrating the necessity of optimizing both modules. In particular,

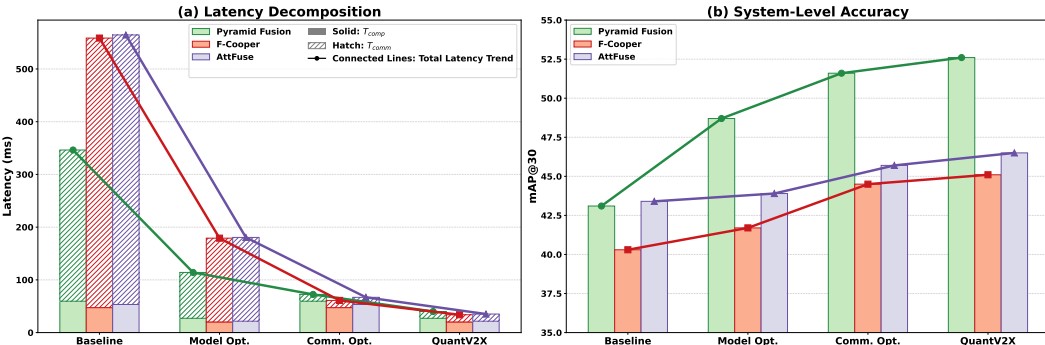

Figure 6: **System-Level Efficiency and Accuracy Decomposition Analysis.** We evaluate four distinct system configurations: *Baseline* (FP32), *Model Opt.* (Quantization only), *Comm. Opt.* (Codebook only), and the fully unified *QuantV2X*. **(a) Latency Decomposition:** The stacked bars decompose total latency into computation ($T_{comp}$) and communication ($T_{comm}$). The trend illustrates the reduction in total latency. Note that Model Opt. specifically lowers the computation floor, while Comm. Opt. alleviates the transmission bottleneck. **(b) System-Level Accuracy:** The corresponding detection performance across three fusion architectures (Pyramid Fusion, F-Cooper, AttFuse). These trends highlight that QuantV2X delivers the highest accuracy with lowest total latency and is generalizable to different fusion architectures.

these trends are observed across all evaluated architectures (Pyramid Fusion Lu et al. (2024), F-Cooper Chen et al. (2019a), AttFuse Xu et al. (2022c)), where QuantV2X delivers the highest accuracy and lowest latency.

## 3.3 SCALING BEHAVIOR OF QUANTV2X UNDER GPU RESOURCE BUDGETS

We examine the scaling behavior of QuantV2X by varying the backbone capacity of both the full-precision baseline and QuantV2X under different GPU memory budgets for common in-vehicle GPUs, as shown in Fig. 7. For each memory budget, we allocate the largest feasible model that can fit within the available resources. Once memory limits are imposed, larger backbones in the full-precision systems cannot be accommodated without downsizing the model, which leads to noticeable performance degradation. In contrast, QuantV2X effectively bridges this gap by compressing larger models into compact low-bit representations that remain within device-level memory constraints while still achieving high perception accuracy. This capability demonstrates that QuantV2X not only alleviates efficiency bottlenecks but also fundamentally enables *scalability under resource constraints*. By unlocking the potential to deploy larger and more accurate cooperative perception models on edge devices, QuantV2X provides a practical pathway to scaling state-of-the-art cooperative perception in real-world resource-constrained settings.

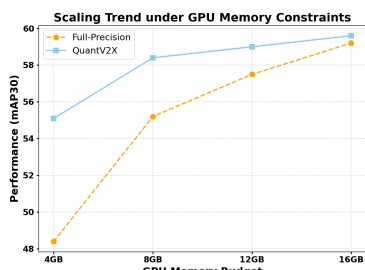

Figure 7: Scaling trend under different GPU memory constraints. QuantV2X enables deployment of larger models under tight budgets while maintaining high perception performance.

## 4 CONCLUSION

In this work, we introduce QuantV2X, a fully quantized multi-agent system designed to tackle the system-level inefficiencies prevalent in cooperative perception. QuantV2X achieves substantial reductions in cumulative system latency and communication overhead while maintaining competitive perception performance relative to an ideal full-precision baselines. Our findings highlight the potential of quantized multi-agent systems as a practical and scalable solution for resource-constrained deployment for V2X cooperative perception. We advocate for reframing cooperative perception research around system-level efficiency, latency, and deployability, a perspective we show is critical for transitioning V2X from research prototypes to scalable real-world deployment. As future directions, we aim to deploy QuantV2X in real-world Cellular-V2X testbeds to conduct comprehensive evaluations under practical deployment conditions.

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

Appendix

## A   Related Work

**Quantization Methods Overview.** Existing quantization techniques can be broadly categorized into two main paradigms: (1) Quantization-Aware Training (QAT) and (2) Post-Training Quantization (PTQ) (Krishnamoorthi, 2018). QAT methods (Esser et al., 2019; Zhuang et al., 2020; Chen et al., 2021) necessitate access to complete labeled training datasets, making them computationally intensive but generally more accurate. In contrast, PTQ offers a more lightweight alternative by enabling quantization using limited unlabeled data, eliminating the need for full retraining and thereby significantly reducing computational overhead. Numerous PTQ techniques have been developed for 2D vision tasks (Wu et al., 2020; Nahshan et al., 2019; Yuan et al., 2021; Li et al., 2021a; Liu et al., 2022; 2021), as well as typically leveraging max-min or entropy-based calibrations for INT8 quantization. Notably, BRECQ (Li et al., 2021a) introduces block-wise reconstruction to refine PTQ accuracy, while PD-Quant (Liu et al., 2022) mitigates overfitting by utilizing batch normalization (BN) statistics to adjust activation distributions. However, directly applying these PTQ strategies to 3D point cloud tasks results in severe performance degradation, as evidenced in LiDAR-PTQ (Zhou et al., 2024a). Moreover, quantization challenges in multi-modal multi-agent cooperative perception systems remain underexplored.

**Cooperative Perception.** Cooperative perception enhances perception and downstream tasks' performance, such as planning and prediction by integrating sensory information across multiple connected agents (Shi et al., 2022; 2023; Wang et al., 2025; 2020; Zhang et al., 2025; Yi et al., 2019; van Hoek et al., 2021; Xie et al., 2024; Lei et al., 2025a; Gao et al., 2025b). Depending on the type of shared data, existing cooperative perception frameworks can be classified into three main paradigms: late fusion, where detection results are exchanged (Rawashdeh and Wang, 2018); early fusion, which involves transmitting raw LiDAR point clouds (Chen et al., 2019b); and intermediate fusion, which has emerged as the dominant approach by striking a balance between accuracy and bandwidth efficiency through the exchange of compressed neural features (Xiang et al., 2023; Xu et al., 2022b;a). Intermediate fusion techniques can be further divided into (1) computation-based fusion and (2) learning-based fusion. For computation-based fusion, F-Cooper (Chen et al., 2019a) employs max pooling to aggregate voxel features in multi-agent scenarios, while AttFuse (Xu et al., 2022c) adopts agent-wise single-head attention for feature integration. In contrast, learning-based methods such as V2X-ViT (Xu et al., 2022b) leverage vision transformers for multi-agent perception, Where2comm (Hu et al., 2022) that leverages a spatial confidence map for communication-efficient collaboration, and Pyramid-Fusion (Lu et al., 2024) applies a multi-scale convolutional network to enhance feature fusion in the bird's-eye view (BEV) space. Despite the growing body of work on fusion strategies, their inherent computational and memory overhead poses a significant challenge, particularly for real-time deployment (Xiang et al., 2025). Previous works on communication-efficient cooperative perception systems (Hu et al., 2024; 2022; Wang et al., 2020; Liu et al., 2020b;a; Yang et al., 2023) primarily focus on reducing communication latency. Instead, we introduce a comprehensive quantized system that improves efficiency from both model inference and transmission.

## B   Motivating Research Questions

**Question:** What's the main research motivation behind QuantV2X?

**Answer:** The primary motivation behind QuantV2X is to investigate whether a quantized intermediate fusion system can effectively replace its full-precision counterpart in cooperative perception. Building on insights from prior work such as V2X-ReaLO (Xiang et al., 2025), full-precision systems have been shown to be deployable in real-world settings, but their inefficiency often leads to significantly degraded performance in practice. QuantV2X is therefore grounded in the principle of *real-world applicability*, aiming to meet key deployment requirements such as low system-level latency, reduced memory footprint, and minimal performance degradation. Through the experiments presented in the main paper, we demonstrate that QuantV2X offers a compelling perspective for shifting the focus from full-precision systems to low-bit alternatives, and we validate its practical relevance with extensive experiments.

**Question:** What's the significance of the real-world applicability of QuantV2X?

**Answer:** QuantV2X aims to resolve the system-level latency bottlenecks presented in current cooperative perception systems. The end-to-end latency of a cooperative perception system can be decomposed into three major components: (i) the time each agent's model needs to process its own sensor data (local inference), (ii) the time it takes to send information between different agents (communication), and (iii) the time needed to process all the received information and output perception results (fusion). In real-world deployments, full-precision models and data create major bottlenecks at all three stages: (i) heavy computation during local inference, (ii) large data sizes that slow down communication between agents, and (iii) limited memory capacity for storing feature buffers. These bottlenecks collectively undermine real-time performance, especially under resource and bandwidth constraints typical of practical V2X deployments. To systematically address these bottlenecks, we propose QuantV2X, a fully quantized cooperative perception system. Full-stack quantization plays a crucial role in improving performance at every stage. First, by quantizing both the perception models and fusion modules, we speed up local inference with faster low-precision computation. Second, by transmitting quantized code indices instead of BEV features in FP32 format, we greatly shrink the communication payload, reducing the time needed to exchange information between agents. Third, the smaller memory footprint of quantized models and feature maps makes it possible to store and manage more historical BEV features within the limited GPU resources to enhance collaboration performance. Our extensive experiments demonstrate that QuantV2X meets the demands of real-world deployment. This is particularly impactful given the limited exploration of quantized systems for cooperative perception in the current literature.

**Question:** What does "fully quantized" mean?

**Answer:** The "fully quantized" means that our quantized cooperative perception system is quantized in an end-to-end manner, from the perception backbone, compressor module, fusion module, and downstream head. Through this design, we aim for ultimate inference speed and communication bandwidth reduction and the lowest memory requirements.

**Question:** Why does the naive quantization method not work in cooperative perception scenarios?

**Answer:** Naive quantization results in a huge amount of precision loss. The challenge of quantization mainly stems from the heterogeneity of different modalities of input, making the activation range vary across different collaborating agents. Besides, the spatial feature is often misaligned. Naive quantization results in a huge amount of information loss and thus degrades the performance badly. Thus, we propose an important alignment module to resolve the above-mentioned challenges. As shown in Fig. 3 in the main paper, our alignment module effectively aligns closely with the full-precision model, resulting in less BEV feature precision loss during the multi-agent fusion stage, fewer false positive detections, and enabling the quantized model to output 3D bounding boxes with more precise coordinates and higher confidence scores.

## C  DISCUSSIONS OF TECHNICAL DESIGNS IN QUANTV2X

**Question:** Why are LLM quantization methods not directly applicable to V2X systems?

**Answer:**

1. **Overview of LLM-based quantization methods:** Large Language Model (LLM) quantization techniques, such as GPTQ (Frantar et al., 2022), AWQ (Lin et al., 2024), and SmoothQuant (Xiao et al., 2023), are primarily designed for autoregressive Transformer architectures operating on discrete token sequences. These methods aim to reduce bit precision while preserving semantic prediction accuracy for language understanding and generation. The optimization strategies typically rely on statistical characteristics of text-based embeddings and the error tolerance inherent to NLP tasks, which do not generalize automatically to other domains.

2. **Mismatch in data input and model architecture:** In V2X systems, the input domain consists of heterogeneous, high-dimensional sensor data (LiDAR point clouds, camera frames, radar signals, and cooperative messages), which are continuous, structured in 3D space, and often fused across modalities. Model architectures for V2X tasks are likewise diverse: voxel/BEV encoders, 3D CNN backbones, sparse convolutional layers, graph-based fusion, and task-specific heads for detection. These differ substantially from the pure

Transformer decoders that dominate LLM design. As a result, quantization error manifests differently, especially in spatial perception features where geometric consistency is critical.

3. **Quantization degradation from direct adoption:** Applying "off-the-shelf" LLM quantization pipelines to V2X models leads to severe accuracy degradation. Unlike language tasks, where minor numerical perturbations may still yield acceptable output, V2X perception and decision-making require high fidelity in feature representation. Small quantization-induced shifts in point cloud features or cooperative fusion tensors can propagate into large deviations in detected object positions or trajectories, jeopardizing safety-critical decisions.

**Question:** How can quantization methods be applied to other recently published LLM-based V2X work?

**Answer:** We discuss the possibility of applying quantization methods to other LLM-based V2X frameworks. From a V2X quantization perspective, LangCoop (Gao et al., 2025a) is primarily centered on images and LLM reasoning. Since it bypasses cooperative perception and instead relies on a camera input together with language-based communication, the main computational bottleneck lies in the vision-language model inference. For this type of framework, quantization would not target perception modules but rather the Large Vision-Language Model (LVLM) itself. Applying quantization, model shrinking, and task-specific fine-tuning can significantly reduce latency and memory usage, which is crucial if LangCoop is to be deployed in real-time cooperative driving scenarios.

On the other hand, CoLMDriver (Liu et al., 2025) is built on top of cooperative perception while engaging in LLM-based negotiation to resolve driving conflicts. Here, quantization plays an important role in the cooperative perception pipeline. Integrating our work can improve both bandwidth efficiency and inference speed. This means that quantization makes the perception sharing both faster and more reliable, which in turn provides stronger inputs for the negotiation module. In this way, CoLMDriver benefits from quantization not by directly accelerating the LLM negotiation but by improving the accuracy and timeliness of cooperative perception.

**Question:** Can we do 2-bit quantization?

**Answer:** We did not incorporate 2-bit quantization into QuantV2X because such extreme precision reduction is impractical for safety-critical V2X cooperative perception, as shown in Table 5. Our experiments already show that even under INT4 weight quantization with INT8 activations, the system requires careful calibration and the proposed alignment module to recover accuracy close to full precision. Moving down to 2-bit precision leads to substantial quantization noise: feature distributions from heterogeneous agents become unstable, and small perturbations in BEV features propagate into large errors in detection and

Table 5: 2-Bit Quantization Performance on DAIR-V2X Dataset (collaboration mode: $L_P + C_R$).

| Method | Bits (W/A) | AP@30 / AP@50 |
|---|---|---|
| | 32 / 32 | 75.1 / 68.2 |
| **Pyramid Fusion** | 8 / 8 | 74.6 / 67.8 |
| | 4 / 8 | 74.2 / 66.7 |
| | 2 / 8 | 40.8 / 37.0 |

fusion. In multi-agent scenarios with sensor misalignment and communication latency, this error amplification becomes unacceptable.

Moreover, 2-bit quantization is not well supported by mainstream inference engines such as TensorRT, making deployment on real-time edge platforms infeasible. Since QuantV2X is explicitly motivated by practical deployment, we focus on 8-bit settings, which balance efficiency and reliability. These settings already reduce system-level latency by ×3.2 while preserving up to 99.8% of full-precision accuracy, demonstrating both feasibility and robustness without resorting to ultra-low bandwidths.

# D ADDITIONAL DETAILS ON MODEL-LEVEL EXPERIMENTS

## D.1 MORE EXPERIMENTAL SETTING

**Implementation details.** We train and evaluate all full-precision models using the open-source HEAL repository (Lu et al., 2024). For each model listed in the main paper, we follow a standardized training protocol of 40 epochs and select the best-performing checkpoint as the full-precision baseline. Post-training quantization (PTQ) is then applied to these selected models. For PTQ, we use 0.5% of

the original training dataset as the calibration set, and perform 5,000 calibration steps. We conduct ablation studies on this calibration setup, with results provided below. All experiments are conducted on an NVIDIA A6000 GPU. Unless otherwise specified, all additional results presented refer to the Pyramid Fusion model (Lu et al., 2024).

## D.2 MORE EXPERIMENT RESULTS

**Ablation Study: Effect of Calibration Dataset Size.** In our main experiments, we use 0.5% of the training dataset as the calibration set during the PTQ stage. In Table 6, we present the impact of varying the calibration dataset size. We observe that using just 0.5% of the training data is already sufficient to achieve strong quantization performance, with a minimal performance drop compared to using larger subsets.

Table 6: Effect of Calibration Dataset Size of QuantV2X in DAIR-V2X dataset. Bits (W/A) is set to INT4/8 and results are displayed in terms of AP30/50.

| Full-Prec. | 0.25% | 0.5% | 1% |
|---|---|---|---|
| 75.1/68.2 | 73.8/66.5 | 74.2/66.7 | 74.3/66.9 |

**Ablation Study: Effect of Calibration Steps.** We also investigate the impact of the number of calibration steps during the PTQ process. As shown in Table 7, we find that 5,000 steps provide effective calibration, and increasing the number of steps beyond this point yields diminishing returns in terms of performance improvement.

Table 7: Effect of Calibration Steps of QuantV2X in DAIR-V2X dataset. Bits (W/A) is set to INT4/8 and results are displayed in AP30/50.

| Full-Prec. | 1000 | 5000 | 20000 |
|---|---|---|---|
| 75.1/68.2 | 73.2/65.7 | 74.2/66.7 | 73.8/65.9 |

**Ablation Study: Comparison with other quantization baselines.** We compare QuantV2X (INT8 W/INT8 A) with low-precision training (FP16). As shown in Table 8, QuantV2X shows better performance and lower latency compared to other baselines.

**Quantization results under V2X-Real and OPV2V datasets.** As shown in Table 9, our PTQ stage leads to a minimal performance drop compared to full-precision baselines across different domains.

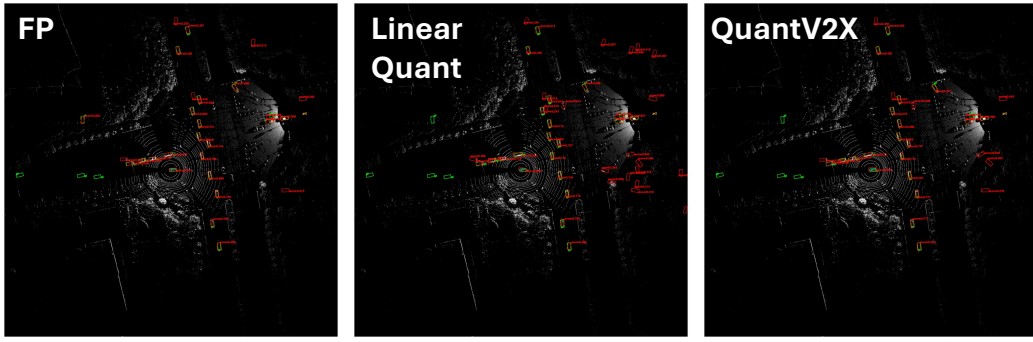

Figure 8: Qualitative results on DAIR-V2X dataset (Collaboration mode: $L_P + C_R$). Green and red bounding boxes denote the ground-truth and predicted detection results, respectively.

Table 8: Comparison with other quantization baselines in V2X-Real dataset.

| | Full-Prec. | Low-Prec. | QuantV2X |
|---|---|---|---|
| mAP30/50 | 53.8/43.5 | 53.0/42.7 | 53.4/43.0 |
| Latency (ms) | 59.5 | 43.5 | 27.1 |

Table 9: Performance of QuantV2X in V2X-Real and OPV2V datasets.

| Bits(W/A) | V2X-Real (mAP30/50) | OPV2V (AP30/50) |
|---|---|---|
| 32/32 | 53.8/43.5 | 97.9/97.1 |
| 4/8 | 52.5/42.8 | 97.6/96.7 |

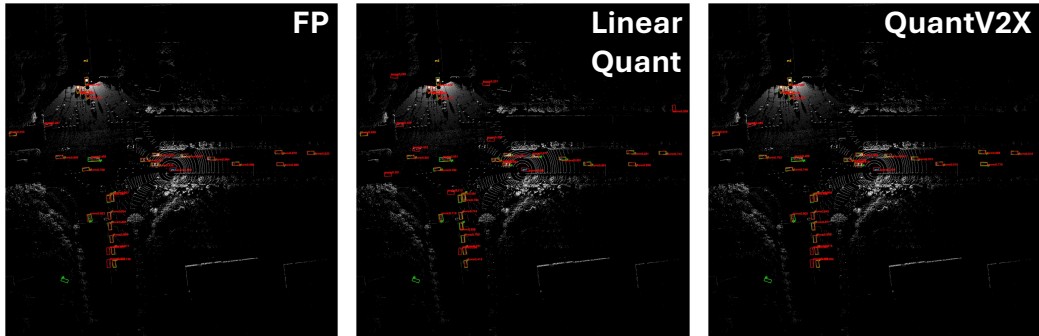

Figure 9: Qualitative results on DAIR-V2X dataset (Collaboration mode: $\mathbf{L_P} + \mathbf{L_S}$). Green and red bounding boxes denote the ground-truth and predicted detection results, respectively.

### D.3 QUANTIZATION EFFECT ON DIFFERENT FUSION METHODS

We demonstrate the generalizability of our PTQ process with different fusion methods in the main paper. In this subsection, we further conduct an analysis of the quantization effect on different fusion methods. Fusion methods in V2X perception can be broadly categorized into the following groups:

1. **Computation-based methods:** These include approaches like AttFuse (Xu et al., 2022c) and F-Cooper (Chen et al., 2019a), which rely on predefined computation (e.g., max or mean fusion) without learnable fusion networks.
2. **CNN-based methods:** Methods such as Pyramid Fusion (Lu et al., 2024) and Who2com (Liu et al., 2020b) utilize convolutional neural networks for feature fusion, enabling learnable fusion strategies.
3. **Attention-based methods:** V2X-ViT (Xu et al., 2022b) and Where2comm (Hu et al., 2022) employ attention mechanisms to model inter-agent relationships.

Table 1 in the main paper summarizes the quantization impact on each of these fusion methods. Among the computation-based methods, AttFuse remains robust under quantization, while F-Cooper suffers notable performance degradation. This is attributed to F-Cooper's max-pooling mechanism, which is sensitive to outliers, and the performance could be exacerbated by precision loss during quantization. In contrast, AttFuse can better preserve interaction cues, even when BEV features are quantized to lower bits.

CNN-based methods like Pyramid Fusion and Who2com demonstrate strong resilience to quantization. This robustness arises from their use of standard convolutional layers, where quantization-aware calibration techniques can effectively align feature distributions.

For attention-based methods, Where2comm shows reasonable performance, likely due to its relatively simple attention structure with fewer layers. On the other hand, V2X-ViT experiences more pronounced degradation. Its architecture includes complex operations like LayerNorm and window-based attention, which are more sensitive to quantization and rely heavily on agent-specific feature interactions. Despite our alignment module, some information loss is inevitable in such deep attention-based pipelines. More advanced quantization strategies tailored to these operations are needed to preserve performance in models like V2X-ViT.

### D.4 DISCUSSION OF ALIGNMENT MODULE IN PTQ STAGE

Although PTQ has shown promising results in single-agent RGB or LiDAR-based perception tasks (Liu et al., 2022; Zhou et al., 2024a), extending it to multi-agent V2X scenarios introduces unique challenges. Unlike the single-agent case, V2X systems involve multiple agents equipped with diverse sensor modalities and observing the environment from varying viewpoints, resulting in inconsistent feature distributions across agents. This cross-agent heterogeneity undermines the

Table 10: Evaluation of alignment module on DAIR-V2X ($L_P + C_R$) decomposing performance by distance (Short: 0-30m, Mid: 30-50m, Long: 50m+) and IoU threshold.

| Bits (W/A) | Config | AP@0.3 | | | AP@0.5 | | | AP@0.7 | | | Total AP | | |
|---|---|---|---|---|---|---|---|---|---|---|---|---|---|
| | | 0-30 | 30-50 | 50+ | 0-30 | 30-50 | 50+ | 0-30 | 30-50 | 50+ | @0.3 | @0.5 | @0.7 |
| FP32 / 32 | Full Prec | 86.8 | 82.1 | 63.3 | 82.2 | 76.1 | 54.9 | 68.5 | 57.9 | 36.7 | 75.1 | 68.2 | 51.0 |
| INT4 / 8 | MinMax | 86.4 | 80.6 | 59.4 | 79.0 | 68.0 | 46.4 | 46.1 | 33.3 | 13.5 | 73.2 | 61.5 | 24.0 |
| INT4 / 8 | + AdaRound | 84.9 | 80.2 | 60.2 | 79.2 | 73.3 | 51.1 | 49.0 | 41.0 | 22.0 | 72.8 | 65.1 | 34.7 |
| INT4 / 8 | + Alignment | **86.0** | **80.8** | **62.5** | **80.8** | **73.9** | **53.7** | **53.3** | **41.4** | **27.9** | **74.2** | **66.7** | **38.3** |

assumptions of standard PTQ methods, which typically neglect the dynamic and inconsistent activation statistics inherent in multi-agent settings. Real-world deployment further exacerbates this issue. Sensor noise, localization drift, and communication latency can introduce spatial misalignment in shared features, resulting in unstable activation ranges. These fluctuations are particularly harmful at low bit precision, where even small shifts can cause significant quantization errors. To mitigate these challenges, we propose a novel alignment module that compensates for both spatial misalignment and feature distribution variation across heterogeneous agents. As illustrated in Fig. 3, our alignment module significantly reduces the quantization-induced degradation and better preserves full-precision feature distribution.

### D.5 DISCUSSION OF THE ROLE OF ADAROUND IN PTQ STAGE

In this section, we provide an in-depth ablation study to demonstrate that AdaRound Nagel et al. (2020) and the Alignment Module are not redundant components; rather, they play distinct, synergistic roles in the QuantV2X framework. Specifically, AdaRound serves as the necessary foundation for weight stability, while the Alignment Module addresses the domain-specific challenges of collaborative perception (CP), particularly regarding long-range precision and global optimization.

**Observation 1: Alignment module solves the "precision & range" challenge (where AdaRound hits a ceiling).** While AdaRound stabilizes the weights, it hits a performance ceiling on strict metrics. As shown in Table 10, AdaRound alone struggles with high-precision localization (AP@0.7) and long-range detection (>50m). The alignment module provides a massive boost: for AP@0.7 (50m+), it improves performance from 22.0 (AdaRound) to 27.9 (Ours), a 27% relative improvement. Those results indicate that the alignment module is quite important in preserving the downstream detection performance in longer-range scenarios with higher requirements of precision.

**Observation 2: AdaRound is necessary to preserve CP system in a locally optimized state, but with alignment module the quantized system is able to reach the globally optimized state.** We notice that naive rounding method (e.g., nearest neighbor) demonstrates that standard nearest-neighbor quantization degrades the performance, as shown in Table 11. Since naive rounding optimizes error locally for each weight based solely on magnitude, this creates a systematic rounding bias (e.g., consistently rounding up) that ignores the layer's global output distribution. In multi-agent scenario, when naive rounding is applied, the systematic bias from every agent compounds during this summation. This makes the noise level of the fused feature map so high that the semantic and geometric feature of the scene is destroyed, which is consequently aligned with our observation that applying the Alignment Module on top of Naive Rounding failed to recover performance. AdaRound on the other hand adapts the rounding to minimize layer-wise reconstruction error. This effectively lowers the rounding error and preserving the basic structural integrity of the model weights and creates a more alignable feature space for multi-agent fusion. However, as more agents collaborate, the compounding errors from different agents make the AdaRound not to best optimized state as it largely focuses on single-agent's local optimization. That's why the alignment module is designed to calibrate the quantized model to a globally optimized state that it does not need to perform perfect layer-wise optimization but just ensures that the key objectives (e.g., heterogeneous feature property and final spatial property, especially the scenarios that require high precision in long-range distance) are well-preserved (as Table 10 supported).

Table 11: Performance of Naive Rounding with Alignment Module on the DAIR-V2X dataset ($L_P + C_R$ configuration).

| Bits (W/A) | Configuration | AP@0.3 | AP@0.5 |
|---|---|---|---|
| FP32 / 32 | Full Precision | 75.1 | 68.2 |
| INT4 / 8 | Max-min Baseline | 73.2 | 61.5 |
| INT4 / 8 | + Naive Rounding | 70.0 | 58.6 |
| **INT4 / 8** | **+ Alignment Module** | **70.3** | **59.3** |

### D.6 MORE QUALITATIVE RESULTS

Fig. 8 and Fig. 9 demonstrates more qualitative results. Note that naive quantization methods lead to many false positive detections, whereas QuantV2X achieves comparable detection capability with the full-precision model.

## E ADDITIONAL DETAILS ON SYSTEM-LEVEL EXPERIMENTS

### E.1 SYSTEM-LEVEL LATENCY MEASUREMENT SETTING

To accurately profile the latency of each model, we first export the models to ONNX format and deploy them using the TensorRT platform (Migacz, 2017). Latency measurements are obtained by averaging the results over 10 runs on an NVIDIA RTX 3090 GPU. Since native TensorRT does not support quantization for certain network modules, we implement custom CUDA kernels and integrate them as TensorRT plug-ins to ensure compatibility and accurate latency profiling. For communication latency, we follow (Xiang et al., 2025) and calculate the latency between the full-precision BEV feature and quantized message representation. The testing is conducted on edge platforms in either vehicle or infrastructure, as illustrated in Fig. 10. Note that V2X-ReaLO (Xiang et al., 2025) is an open-source, ROS-based framework and dataset designed to deploy and evaluate cooperative perception algorithms on real-world vehicles and smart infrastructure. Unlike static benchmarks, it facilitates the online execution of intermediate fusion pipelines, enabling the rigorous validation of deployment-critical metrics under dynamic physical constraints.

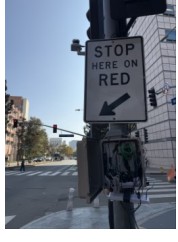
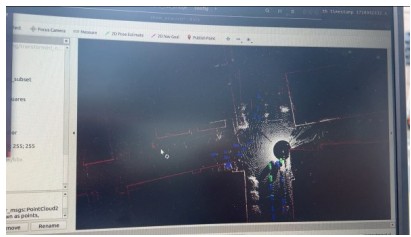
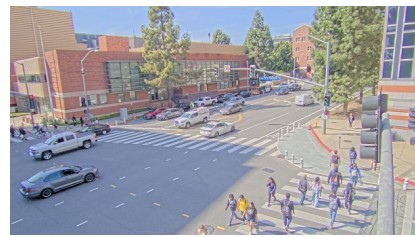
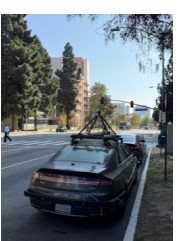
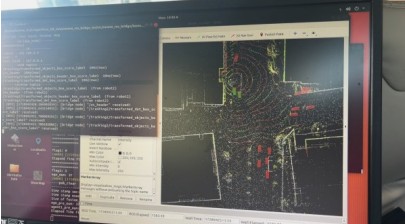
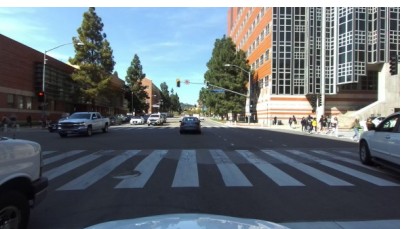

Figure 10: Illustration of real-world testing platform. *Upper*: illustration of infrastructure-side edge testing platform. *Lower*: illustration of vehicle-side edge testing platform.

Table 12: Power consumption comparison.

| Precision | Power (W) | Throughput (QPS) | Energy / Query (J) | Efficiency (QPS/W) |
|---|---|---|---|---|
| FP32 | 330 | 47.6 | 7.02 | 0.144 |
| INT8 | 300 | 124 | 2.41 | 0.413 |
| *INT8 vs. FP32 gains:* Speedup $= 2.61\times$, Energy $\downarrow 65.7\%$, Efficiency $= 2.87\times$ | | | | |

Table 13: Ablation on the impact of $n_L$ and $n_R$ selection of QuantV2X in V2X-Real dataset. Message size is reported in megabytes (MB). We report both the ideal accuracy (without latency considerations) and the system-level accuracy (mAP30/50).

| $n_L$ | $n_R$ | Message Size (MB) | Ideal Acc. | System Acc. |
|---|---|---|---|---|
| 16 | 1 | 0.016 | 49.9/40.6 | 49.6/39.6 |
| 16 | 2 | 0.033 | 51.8/42.1 | 51.4/40.9 |
| 32 | 1 | 0.021 | 51.2/41.5 | 50.8/40.5 |
| 32 | 2 | 0.042 | 51.4/41.8 | 50.7/40.4 |
| 64 | 1 | 0.025 | 51.9/41.4 | 51.3/40.3 |
| 64 | 2 | 0.050 | 52.1/42.3 | 51.9/41.3 |
| 128 | 1 | 0.029 | 53.2/43.0 | 52.6/42.2 |
| 128 | 2 | 0.059 | 53.6/43.6 | 52.4/41.5 |
| 256 | 1 | 0.034 | 52.7/43.1 | 52.2/41.7 |
| 256 | 2 | 0.067 | 52.5/42.4 | 51.8/41.0 |

### E.2 POWER CONSUMPTION MEASUREMENT

Table 12 reports power and throughput comparison between FP32 and INT8 inference on NVIDIA RTX 3090. Energy per query (J) is computed from the reported power and throughput, and efficiency is expressed in QPS/W. Relative gains of INT8 over FP32 are also provided. The measurement follows prior work (Han et al., 2015; Wang et al., 2019; Desislavov et al., 2023) that optimizes quantization and measures efficiency.

### E.3 ADDITIONAL DETAILS ON CODEBOOK LEARNING

In the codebook learning stage, we first pretrain the codebook for 20 epochs with codebook parameters updated exclusively. After this stage, we perform joint training of the entire system for an additional 10 epochs and select the best model based on validation accuracy. Table 13 reports an ablation study on the effect of $n_L$ and $n_R$. For all experiments in the main paper, we adopt the configuration that achieves the highest system-level perception accuracy.

### E.4 DIFFERENT FUSION MODELS LATENCY MEASUREMENTS

Table 14 presents the model-level latency (local latency + fusion latency) of each fusion method. Compared with other methods, Pyramid Fusion achieves the best perception performance while maintaining a competitively low latency.

Table 14: Model-level Latency (ms) across different fusion methods.

| Bits (W/A) | Pyramid Fusion (Lu et al., 2024) | F-Cooper (Chen et al., 2019a) | AttFuse (Xu et al., 2022c) | V2X-ViT (Xu et al., 2022b) | Who2com (Liu et al., 2020b) | Where2comm (Hu et al., 2022) |
|---|---|---|---|---|---|---|
| 32/32 | 59.5 | 53.3 | 47.4 | 102.4 | 64.6 | 44.3 |

### E.5 DISCUSSION OF QUANTIZED SYSTEM IN REAL-WORLD SCENARIOS

In practical V2X deployments, each stage of the cooperative perception pipeline introduces substantial challenges. First, full-precision deep neural networks are computationally expensive and poorly suited for low-power edge devices, resulting in slow local inference. Second, the high dimensionality of BEV (Bird's Eye View) feature maps, typically represented in 32-bit floating-point format (FP32), leads to significant communication overhead, making timely feature exchange between agents difficult.

Third, memory constraints on edge devices restrict the number of BEV feature buffers that can be retained, increasing the likelihood of missed or delayed information exchange across agents.

QuantV2X systematically addresses these bottlenecks in real-world scenarios through full-stack quantization.

1. **Model-side efficiency.** By quantizing both the perception models and fusion modules, QuantV2X accelerates both local and fusion inference using lightweight low-precision (INT8) computation.

2. **Transmission efficiency.** By transmitting low-bit quantized codebook indices instead of FP32 BEV feature, the communication payload is greatly reduced, lowering transmission latency and enabling more timely collaboration.

3. **Memory efficiency.** The reduced memory footprint of quantized models and feature maps allows for storing and managing a greater number of historical BEV features within limited GPU resources, improving the temporal richness of collaborative data.

In resource-constrained environments, many informative collaborative cues are often neglected due to latency, memory, or bandwidth limitations in full-precision systems. QuantV2X mitigates these issues, enabling more effective use of collaborative information and reducing performance degradation. These improvements explain the consistent performance gains observed in real-world evaluation scenarios.

## F  BROADER IMPACT

In addition to the research contributions presented in this paper, our work provides significant engineering value for real-world practicality to the community. We will open-source our findings to help advance V2X research in the context of model quantization. Specifically, we have enhanced the original HEAL codebase to be more deployment-friendly and have integrated it with the real-world testing platform V2X-ReaLO. This bridges the gap between software development and hardware-level optimization, enabling practical deployment of quantized V2X perception models. To the best of our knowledge, this is the first exploration of full-stack quantization in the V2X domain. We believe our ecosystem will have a substantial impact by laying the groundwork for future research and fostering broader discussions in the community.

## G  LLM USAGE

In preparing this manuscript, Large Language Models (LLMs) were employed strictly as writing assistants. Their use was limited to improving grammar, clarity, and stylistic polish of the text. No LLM was involved in formulating research ideas, designing or conducting experiments, analyzing data, or drawing scientific conclusions.

