# OpenReview forum: "QuantV2X: A Fully Quantized Multi-Agent System for Cooperative Perception"
_ICLR.cc/2026/Conference — Submitted to ICLR 2026_

### Official Review · Reviewer_HTY5 · 2025-10-30

**Soundness:** 3
**Presentation:** 2
**Contribution:** 3
**Rating:** 6
**Confidence:** 3

**Summary:**

This paper presents QuantV2X, a fully quantized end-to-end cooperative perception framework for Vehicle-to-Everything (V2X) systems. It introduces unified quantization across both neural models and transmitted feature representations, aiming to reduce computation, communication bandwidth, and latency without compromising perception accuracy.

**Strengths:**

- The focus on system-level deployability rather than only model accuracy is refreshing and directly relevant to real-world autonomous driving. QuantV2X targets both computation and communication bottlenecks, which are typically overlooked in prior V2X works.
- The integration of quantization into both the perception network and inter-agent communication pipeline is technically comprehensive and well-justified. The unified INT4/INT8 pipeline and compatibility with TensorRT suggest good deployment readiness.
- Introducing heterogeneity and spatial alignment losses to counter quantization noise is a reasonable solution to a domain-specific problem of feature misalignment in multi-agent fusion.

**Weaknesses:**

- While the integration is well-engineered, the quantization techniques themselves (post-training quantization, AdaRound, codebook compression) are based on existing literature. The main novelty lies in application-level integration. The paper would benefit from a clearer description of what is technically novel beyond combining known quantization and compression techniques.
- The claim that latency reduction can outweigh quantization-induced accuracy loss is convincing, but more quantitative trade-off analysis would strengthen the argument.

**Questions:**

Although component analysis is included, which modules are most sensitive to quantization?

---

> ### Author Response · Authors · 2025-11-20
> **Response to Weaknesses**
>
> We sincerely thank the reviewer for their positive assessment. We are particularly encouraged that they found our "focus on system-level deployability... refreshing and directly relevant" and our technical integration "comprehensive and well-justified."
>
> ## 1. Response to Technical Novelty
> We acknowledge that while the individual quantization primitives (e.g., PTQ, Codebook) are built upon existing literature, our core novelty lies in the non-trivial system-level co-design (in both model-level and communication-level) and domain-specific adaptations (in resolving the domain-specific problem of feature misalignment in multi-agent fusion) required to orchestrate them within a complex, end-to-end collaborative perception pipeline. By successfully constructing the first fully quantized V2X system, QuantV2X moves beyond theoretical simulation to demonstrate practical viability in real-world scenarios. More importantly, we advocate for reframing cooperative perception research around system-level efficiency, latency, and deployability, a perspective we show is critical for transitioning V2X from research prototypes to scalable real-world deployment.
>
> ## 2. Additional Experiments on Quantitative Trade-off Analysis
> We provide the following experiments on decomposition of latency components to analyze the effect of communication-level and model-level inefficiencies on the system-level performance. We report the inference time and their corresponding system-level performance as below.
>
> ### Table A: System-Level Latency Decomposition (ms)
> *Note: $T_{comp} = T_{loc} + T_{fusion}$ (Computation Latency)*
>
> | Method  | Pyramid ($T_{comm}$ / $T_{comp}$) | F-Cooper ($T_{comm}$ / $T_{comp}$) | AttFuse ($T_{comm}$ / $T_{comp}$) |
> | :--- | :--- | :--- | :--- |
> | **Full-Precision (FP)** | 286.8 / 59.5 | 511.2 / 47.4 | 511.2 / 53.3 |
> | **FP + x16 Comp.** | 68.2 / 59.5 | 121.7 / 47.4 | 121.7 / 53.3 |
> | **Codebook Only** | 12.9 / 59.5 | 13.6 / 47.4 | 13.6 / 53.3 |
> | **Quantization Only** | 87.0 / **27.1** | 158.9 / **20.2** | 158.9 / **21.6** |
> | **QuantV2X (Full)** | **12.9** / **27.1** | **13.6** / **20.2** | **13.6** / **21.6** |
>
> ### Table B: System-Level Accuracy (mAP@30 / mAP@50)
> *Note: "Ideal Accuracy" refers to ideal performance with zero latency and no feature compression.*
>
> | Config | Pyramid Fusion | F-Cooper | AttFuse |
> | :--- | :--- | :--- | :--- |
> | **Ideal Accuracy** | 53.8 / 43.5 | 47.3 / 37.9 | 48.3 / 38.5 |
> | **Full-Precision (FP)** | 43.1 / 34.8 | 40.3 / 31.8 | 43.4 / 33.9 |
> | **FP + x16 Comp.** | 48.8 / 38.0 | 41.4 / 32.2 | 44.6 / 35.6 |
> | **Codebook Only** | 51.6 / 41.9 | 44.5 / 33.0 | 45.7 / 34.8 |
> | **Quantization Only** | 48.7 / 38.3 | 41.7 / 32.0 | 43.9 / 34.1 |
> | **QuantV2X (Full)** | **52.6 / 42.2** | **45.1 / 35.0** | **46.5 / 35.9** |
>
> **Key Observations and Insights**:
>
> **1. Insight on communication-level efficiency (impact of codebook)**: Comparing "Full-Precision" to "Codebook Only," we observe that drastically reducing $T_{comm}$ (e.g., 286ms $\rightarrow$ 12.9ms for Pyramid) yields the single largest increase in accuracy (+8.5 mAP@30). This confirms that data freshness is paramount in V2X. Furthermore, the codebook module is also more effective than the standard x16 compression method. For Pyramid Fusion, x16 Compression yields 48.8 mAP@30 with a 68.2ms transmission time. In contrast, our Codebook method yields 51.6 mAP@30 with only 12.9ms transmission time. This demonstrates that learning a discrete codebook is far more efficient for V2X feature transmission than standard compression algorithms.
>
> **2. Insight on model-level efficiency (impact of quantization):** While "Codebook Only" improves results significantly, it still hits a ceiling because the computation time ($T_{comp}$) remains high (~60ms). By adding "Quantization" (our full pipeline), we cut $T_{comp}$ by roughly 55% (59.5ms $\rightarrow$ 27.1ms). This combined reduction allows QuantV2X to achieve an mAP of 52.6, which is remarkably close to the theoretical "Ideal Accuracy" of 53.8, proving that both modules are essential for a fully optimized system.
>
> **3. Generalization across different fusion architectures**:
> Crucially, these observations are consistent across all three fusion methods (Pyramid, F-Cooper, AttFuse). For every architecture, the "Full QuantV2X" setting provides the highest accuracy and lowest total latency, validating that our framework is a generalizable solution for the domain, not specific to one model.
>
> We will integrate these decomposed results and insights into the main paper to clearly illustrate how QuantV2X minimizes both transmission and computation overheads to maximize real-world performance.

---

> ### Author Response · Authors · 2025-11-20
> **Response to question**
>
> In the early stage of the project, to identify the cooperative system's critical bottlenecks when applying quantization, we conducted a preliminary sensitivity analysis by independently quantizing each module. We observed that while individual agent encoders exhibited only moderate degradation under naive PTQ, the multi-agent fusion module suffered a catastrophic drop in performance, identifying it as the most sensitive component. This extreme vulnerability arises because the fusion process relies on precise feature alignment; quantization-induced errors act as noise perturbations that compound and amplify when combining features from multiple agents, significantly exacerbating the inherent feature heterogeneity and spatial misalignments (as detailed in Sec. 2.4.3). This finding was the primary motivation for our Alignment Module: we specifically designed this module with heterogeneity and spatial alignment losses to counteract this amplification effect, forcing the quantized fusion model to mimic the robust behavior of the full-precision baseline and ensuring the high performance recovery we achieved.
>
> We will clarify this finding in the main paper to better motivate our methodological design. We thank the reviewer again for their supportive feedback.

---

### Official Review · Reviewer_cVYj · 2025-10-31

**Soundness:** 3
**Presentation:** 3
**Contribution:** 3
**Rating:** 6
**Confidence:** 3

**Summary:**

This paper presents QuantV2X, the first fully quantized multi-agent system for efficient and scalable V2X cooperative perception. It applies unified end-to-end quantization: PTQ converts pre-trained FP32 models to low-bit networks with spatial alignment correction, while quantized codebook indices replace FP32 BEV features to reduce bandwidth. Experiments on DAIR-V2X, V2X-Real, and OPV2V show QuantV2X retains up to 99.8% accuracy (INT4/INT8), cuts latency by 3.2×, and improves mAP30 by 9.5% under strict GPU memory limits.

**Strengths:**

1. Innovative gap-filling design: It is the first fully quantized system for V2X cooperative perception, shifting from accuracy-centric to system-level efficiency optimization (latency, memory, bandwidth), addressing long-neglected deployment bottlenecks in prior work.

2. Effective technical framework: The PTQ process uses only 0.5% training data for calibration (no full retraining), and the alignment module boosts performance recovery (e.g., from 95.2% to 99.8% for $L_{P}+L_{S}$ in AP30). Codebook-based communication reduces transmission size from 8.6 MB (FP32) to 0.03 MB while preserving feature fidelity.

3. Strong engineering value: Compatible with TensorRT, it achieves 2.61× throughput and 2.87× efficiency gains (INT8 vs. FP32) on RTX 3090.

**Weaknesses:**

1. Limited technical scope: It only validates INT4/INT8 and INT8/INT8 quantization, with no experimental data to support the claim that 2-bit quantization is impractical for V2X. It also focuses on LiDAR-camera fusion but ignores other modalities (radar), limiting applicability to complex multi-modal scenarios.

2. Narrow experimental design: System latency tests rely on fixed ROS/TensorRT environments, excluding real-world variables (signal occlusion, diverse on-board chips). No long-term stability or extreme environment (bad weather) tests are conducted, and comparisons lack state-of-the-art LLM-derived quantization methods (e.g., improved GPTQ).

**Questions:**

See the weaknesses.

---

> ### Author Response · Authors · 2025-11-19
>
> We are deeply grateful to the reviewer for their positive evaluation and for providing an excellent summary of our work. We are encouraged that they recognized our contribution as the "first fully quantized system for V2X cooperative perception" that shifts the focus to "system-level efficiency optimization," and that they found our framework "effective" and of "strong engineering value."
>
> We appreciate the insightful suggestions for extending our work, which highlight important future directions. We address the noted weaknesses below.
>
> ### 1. Response to Limited technical scope
>
> **2-bit Quantization Claim**: We apologize for the lack of clarity. Our claim that 2-bit quantization is "impractical" was based on preliminary experiments where applying standard 2-bit methods resulted in a catastrophic drop in mAP (as shown in the table below), which we deemed unusable for a safety-critical perception system. This severe degradation, especially in the complex fusion module, led us to focus our efforts on the INT4/INT8 regime. We will revise this claim and add this context to the appendix Sec. C in the final version.
>
> *Table: Quantization Performance on DAIR-V2X dataset*
> | Method | Bits (W/A) | AP@30 / AP@50 |
> | :--- | :--- | :--- |
> | **Pyramid Fusion** | 32 / 32 (FP32) | 75.1 / 68.2 |
> | | 8 / 8 | 74.6 / 67.8 |
> | | 4 / 8 | 74.2 / 66.7 |
> | | 2 / 8 | 40.8 / 37.0 |
>
> **Extension to Radar Modality**: This is an excellent suggestion. Our framework was developed using LiDAR-camera fusion, which is itself a common and challenging multi-modal scenario. We chose this focus as it aligns with the SOTA baselines we build upon (e.g., Pyramid Fusion). We strongly agree that extending QuantV2X to include other modalities like radar is a valuable and exciting direction for future work. We believe our framework provides a solid foundation for such extensions.
>
> ### 2. Response to Narrow experimental design
> **Environment setup**: We agree with the reviewer that validating performance under other dynamic real-world variables such as signal occlusion, adverse weather, and hardware heterogeneity, is critical for future deployment. In the meanwhile, we also respectfully argue that our system-level evaluation, by **reporting actual measured latency and throughput on ROS/TensorRT environments, represents a significant advancement over prior collaborative perception works, which often rely primarily on theoretical proxies like FLOPs or parameter counts**. Prior to this work, running cooperative perception models on edge-constrained devices in real-time was computationally infeasible. By establishing the viability of the full quantization pipeline on standard hardware, we lay the necessary groundwork for downstream edge optimization. We will incorporate those valuable suggestions and incorporate them into our future work section.
>
> **Missing SOTA LLM-derived Quantization (e.g., GPTQ)**: We thank the reviewer for this suggestion. Methods like GPTQ have shown outstanding results, primarily in the domain of Large Language Models (LLMs). To our knowledge, the direct application or adaptation of GPTQ to complex, multi-stage collaborative perception BEV-fusion models has not been established. We illustrate the challenges as below:
>
> *Large Language Model (LLM) quantization techniques (GPTQ, AWQ, SmoothQuant, etc.) are primarily designed for autoregressive Transformer architectures operating on discrete token sequences. These methods aim to reduce bit precision while preserving semantic prediction accuracy for language understanding and generation. The optimization strategies typically rely on statistical characteristics of text-based embeddings and the error tolerance inherent to NLP tasks, which do not generalize automatically to other domains. There exists a  mismatch in data input and model architecture from the NLP domain to the V2X domain. In V2X systems, the input domain consists of heterogeneous, high-dimensional sensor data (LiDAR point clouds, camera frames, radar signals, and cooperative messages), which are continuous, structured in 3D space, and often fused across modalities. Model architectures for V2X tasks are likewise diverse: voxel/BEV encoders, 3D CNN backbones, sparse convolutional layers, graph-based fusion, and task-specific heads for detection. These differ substantially from the pure Transformer decoders that dominate LLM design. As a result, quantization error manifests differently, especially in spatial perception features where geometric consistency is critical.*
>
> We will refine such discussion of these emerging LLM-based quantization methods in the appendix Sec. C and note their potential for future investigation in the V2X domain.
>
> We thank the reviewer again for their supportive and constructive review and will incorporate this feedback to improve the clarity of our final paper. We hope these clarifications help better convey the significance and contributions of our work.

---

### Official Review · Reviewer_d6fb · 2025-10-31

**Soundness:** 3
**Presentation:** 2
**Contribution:** 2
**Rating:** 4
**Confidence:** 4

**Summary:**

This paper presents QuantV2X, a fully quantized multi-agent cooperative perception framework aimed at enabling efficient, scalable, and deployable V2X systems. The approach unifies quantization across both neural network models and transmitted BEV feature representations to reduce computational and communication costs.

**Strengths:**

This paper focuses on addressing the critical challenge of deploying collaborative perception systems in realistic environments.
By quantizing both the full-precision model and the shared BEV features, it effectively improves computational and communication efficiency while maintaining—or in some cases slightly improving—accuracy.
Moreover, the system-level experiments, which incorporate realistic processing pipelines, convincingly demonstrate the proposed method’s practicality and its ability to tackle the original motivation.

**Weaknesses:**

- Major Weaknesses
1. Limited methodological novelty despite a strong motivation

The proposed codebook learning stage and PTQ stage constitute the core phases where quantization is performed.
In the codebook learning stage, the authors adopt an existing codebook-based quantization strategy and apply it to BEV feature sharing, rather than introducing a novel technique.
In the PTQ stage, a post-training quantization process is applied for calibration in the collaborative model; however, this approach is a standard practice in the quantization field and does not present any distinctive methodological innovation.
The use of reconstruction loss between quantized and full-precision outputs to align the quantized model follows standard practice in prior quantization studies and lacks novelty.

More critically, the PTQ stage appears to be a direct application of conventional model quantization techniques to the collaborative perception domain, rather than a quantization method specifically designed for multi-agent scenarios.
The stage does not effectively exploit information unique to collaborative perception—such as inter-agent feature dependencies or fusion-related cues that arise during intermediate fusion.
Consequently, aside from the use of codebooks for BEV feature sharing, the proposed PTQ stage could be applied identically to a standalone vehicle model, suggesting that the approach remains a general model-quantization framework rather than a collaboration-aware one.

Furthermore, the quantization procedure itself lacks originality, as it adopts the AdaRound method without modification, and the experiments indicate that AdaRound primarily drives the observed performance gains.

Therefore, although the paper emphasizes being the first to quantize not only BEV features but also the entire collaborative perception model, this contribution appears to be more of an application of existing quantization techniques to the collaborative perception domain rather than a genuinely novel quantization approach that exploits characteristics unique to collaborative perception itself.

2. Concerns about experimental validity

The paper discusses heterogeneity alignment and claims to employ such a setting in its experiments.
However, prior works addressing heterogeneity typically validate alignment by performing inference with auxiliary agents whose sensor–encoder pairs were not used during training.
In contrast, this paper does not specify which sensor–encoder pairs were used in the training and inference stages.
Furthermore, if the same pairs were employed in both, the resulting model would likely be overfitted to those specific combinations.
Hence, without evaluating the model on unseen sensor–encoder pairs, it is difficult to assert that genuine heterogeneity alignment has been achieved.

Additionally, in the system-level experiments (Table 3), two major issues arise.

(1) Unclear decomposition of latency components

In Table 3, the transmission feature size provides only an estimate of how much communication-level inefficiency contributes to $T_{comm}$, whereas the latency associated with model-level inefficiency—$T_{local}$ + $T_{fus}$—cannot be inferred from the reported data.
Consequently, it is difficult to disentangle how performance degrades as a function of $T_{comm}$ and $T_{local}$ + $T_{fus}$.
Reporting inference time and mAP@30/50 for three configurations—(a) after the codebook-learning stage ($T_{comm}$), (b) after the PTQ stage ($T_{local}$ + $T_{comm}$ + $T_{fus}$), and (c) after the PTQ stage but with FP32 BEV feature sharing ($T_{local}$ + $T_{fus}$)—would provide a clearer understanding of performance degradation attributable to each inefficiency component.

(2) Questionable generalization of quantization effects

The degree of accuracy degradation under system-level latency likely depends on the fusion model used.
However, the experiments only compare quantized vs. non-quantized settings for Pyramid Fusion.
To support claims of generality, results should be reported for all fusion models across the same three experimental settings (a–c above), including both inference time and mAP@30/50.

- Minor Weaknesses
1. Insufficient explanation of how the combination weight $\alpha_n$ is generated (page 4).
2. The specific values of $n_L$ and $n_R$ used in QuantV2X are not stated.
3. Poor visual distinction between solid and dashed lines in Figure 4’s legend.
4. Missing explanation for the “Bits (W/A)” notation in Figure 4.
5. The distribution of pose errors is not described.
6. Lack of explanation for the model used as the upper bound.
7. Missing details on BEV feature dimensions (e.g., ego detection range, voxel size).
8. The description of V2X-ReaLO is absent from the main text.
9. The paper reports mAP as the evaluation metric on the V2X-ReaL dataset, but it does not specify which object classes were included in the detection task.

**Questions:**

- Major Questions
1. What methodological novelty justifies this work meeting the acceptance threshold?
2. How are the training and inference settings for heterogeneity alignment designed, and how can the current results support the claim that alignment was achieved?

- Minor Questions
1. Why were KL Divergence and L2 loss chosen for heterogeneity and spatial alignment losses, respectively?
2. For the spatial alignment loss, the paper mentions bounding box representations, but the code uses only the regression head output of the multi-class detection head—why?
3. Why is a uniform distribution used to model communication latency?
4. What are the training times for Stage 2 and Stage 3?

---

> ### Author Response · Authors · 2025-11-20
> **Response to Major Questions**
>
> We sincerely thank Reviewer d6fb for the detailed, constructive feedback and for acknowledging our work's strong motivation and practical, system-level contributions. We are encouraged that the reviewer found our experiments "convincingly demonstrate the proposed method’s practicality."
> We address the reviewer's concerns and questions below.
>
> ### 1. On Methodological Novelty
> **Real-world cooperative perception system suffers from the challenges that the performance of successful collaboration depends on *real-time* *heterogenous* multi-agent collaborations**. Our QuantV2X directly tackles the *real-time* via a unified quantization framework and *heterogeneity* via well-curated calibration procedure. We agree with the reviewer that the individual quantization primitives we use (e.g., codebooks, PTQ, AdaRound) are based on established methods. However, we respectfully clarify that our core methodological innovation lies not in inventing a new atomic quantization algorithm, but in the **novel system-level co-design that tailors and adapts these techniques to solve the aforementioned unique, non-trivial challenges of quantizing an end-to-end collaborative perception (CP) system.**
>
> The reviewer suggests our PTQ stage is a general method that is not "collaboration-aware" and "could be applied identically to a standalone vehicle model." We respectfully disagree and highlight two key collaboration-specific innovations that were motivated by our empirical analysis:
>
> **Fusion-Aware Alignment Module**: A core challenge in CP, which does not exist in standalone models, is the fusion of features from heterogeneous agents. Our initial analysis revealed that a "naive" application of standard PTQ fails catastrophically at this multi-agent fusion stage. This is precisely the "fusion-related cue" the reviewer mentions. Our alignment module was designed specifically to address this. It forces the quantized fusion module to preserve the original system's heterogeneity fusion design, which is a clear methodological adaptation for a multi-agent scenario.
>
> **Multi-Agent-Aware Calibration Procedure**: The reviewer correctly notes that our method would not be "applied identically to a standalone vehicle model." A critical component of our PTQ pipeline (Algorithm 1) is the use of multi-agent aware sampling for the calibration dataset. This ensures the quantized model is robust to a varying number of collaborating agents and different inter-agent dependencies—a concern that is fundamentally unique to collaborative, not standalone, perception.
>
> Moreover, regarding the role of AdaRound, while it is an effective tool, our contribution is not simply "AdaRound without modification." We value the integration of AdaRound in the calibration procedure. The reviewer posits that AdaRound "primarily drives the observed performance gains," but we argue our alignment module is non-trivial and equally critical by observing that it enables the system to recover 99.8% of the full-precision performance even after aggressive quantization as shown in the component analysis.
>
> In summary, we position that QuantV2X is tailored specifically for multi-agent scenarios that resolve the deployment-oriented metrics that are often less discussed in previous literature, and its sophisticated system design effectively meets the requirements and resolves the challenges of *real-time* *heterogenous* multi-agent collaborations.
>
> *However, we respect the research taste of each independent reviewer and we won't forcefully argue that our methodological novelty justifies this work meeting the acceptance threshold.*
>
> ### 2. How are the training and inference settings for heterogeneity alignment designed, and how can the current results support the claim that alignment was achieved?
> We respectfully clarify our definition of "heterogeneity alignment." The reviewer correctly notes that some works define this as generalizing to unseen sensor-encoder pairs.
> In our work, "heterogeneity" refers to the realistic scenario where different agents in the same scene possess different (but known) sensor-encoder setups. The "alignment" challenge we solve is ensuring that our quantized fusion model can effectively process these diverse feature maps without degradation.
> Our experiments (using the same pairs in training and inference, as the reviewer suspected) are designed to validate this specific claim. The results in Table 2 and Figure 3 demonstrate that our alignment module successfully forces the quantized model to recover the full-precision model's behavior, maintaining minimal degradation even when fusing features from these disparate sources. This confirms that our method successfully aligns the quantized representation, preventing the catastrophic failures we observed in naive quantization.
>
> We agree that testing on unseen sensor pairs is an excellent and challenging extension. We will clarify this scope in the final version and add this as a promising direction for future work.

---

> > ### Comment · Reviewer_d6fb · 2025-11-21
> >
> > Thank you for the detailed responses. Many of my questions have been clarified, and the planned revisions will certainly improve readability.
> >
> > However, the response regarding the lack of methodological novelty—my primary concern—was not sufficiently convincing for me to revise my score.
> >
> > First, regarding the Fusion-Aware Alignment Module, I understand that the goal is to align the quantized model with the full-precision model to preserve performance. However, this design essentially follows the knowledge distillation paradigm, where the full-precision model acts as the teacher and the quantized model acts as the student, aligning intermediate representations through reconstruction-based objectives. This appears to be an application of ideas similar to FitNet [1] rather than a fundamentally new, collaboration-aware approach.
> >
> > Second, concerning the Multi-Agent-Aware Calibration Procedure, I believe this procedure is not inherently specific to the collaborative perception domain. If applied to another domain, it would still operate in the same manner as long as the calibration data is sampled from that domain. In this sense, the procedure seems dataset-dependent rather than collaboration-dependent, and therefore does not demonstrate domain-specific novelty.
> >
> > Taken together, I believe the proposed Multi-Agent-Aware Calibration Procedure—including the alignment module—can be applied to any general deep learning model, not only collaborative perception. Any model could be decomposed into modules where outputs are aligned with reconstruction losses on shared inputs. Thus, I do not view the proposed approach as fundamentally "collaboration-aware," and its novelty remains limited.
> >
> > Additionally, based on the ablations presented, most performance improvement appears to come from applying AdaRound. If the alignment module is a key contribution, I strongly recommend including results for a setting where AdaRound is removed (i.e., baseline rounding + alignment module). Such an experiment would more clearly isolate the contribution of the proposed method.
> >
> > Regarding heterogeneity alignment, if the paper’s contribution emphasizes practical deployment, I would expect the heterogeneity setup to reflect realistic and challenging scenarios. Using identical sensor–encoder pairs during training and inference assumes that real-world deployment will not involve unseen sensor configurations, which is unlikely. Despite emphasizing practicality, the current evaluation appears to address only an easier case, which weakens the practical relevance of the heterogeneity claim.
> >
> > Overall, the paper addresses an important limitation in current collaborative perception research, and the direction is highly valuable. However, the strengths of the work are overshadowed by insufficient methodological novelty. With stronger methodological contributions, the work could have a much greater impact.
> >
> > For now, I will maintain my original score, but I welcome further discussion and additional clarification.
> >
> > [1] Romero, Adriana, et al. "Fitnets: Hints for thin deep nets. arXiv 2014." arXiv preprint arXiv:1412.6550 (2014).

---

> > > ### Author Response · Authors · 2025-11-24
> > > **Response to Multi-Agent-Aware Calibration Procedure**
> > >
> > > We sincerely thank the reviewer for the continued engagement and the opportunity to clarify our contributions. We understand that your primary concerns center on the distinction between "application of existing techniques" vs. "domain-specific methodological novelty". We respectfully present the following responses to clarify why our design choices, even those leveraging established primitives, are novel methods / use cases that solve unique challenges in Collaborative Perception (CP) systems.
> > >
> > > ### Response to the novelty of alignment module
> > >
> > > We respectfully argue that the collaborative perception context fundamentally alters the underlying optimization challenge. In single-agent quantization, errors remain local to the specific model. In contrast, collaborative perception introduces a unique bottleneck that quantization noise from $N$ heterogeneous agents does not merely sum up, but compounds non-linearly during feature aggregation due to their different spatialities and input sensor modalities. Consequently, prior methods that optimize modules locally fail to preserve global system performance, as they cannot account for these interaction-driven distortions. Our novelty lies in identifying the fusion stage as this critical failure point and designing the alignment module to specifically target the resulting heterogeneity and spatial misalignment. As the results in main paper demonstrate, where standard quantization fails to prevent feature collapse, our collaboration-aware alignment successfully stabilizes the fusion process. We therefore posit that this is not merely an application of standard calibration to the CP system, but a necessary system-level innovation required to make quantized collaborative perception viable.
> > >
> > > ### Response to the sampling in calibration procedure:
> > > We respectfully posit that this procedure differs fundamentally from general dataset-dependent sampling because it addresses the topological complexity unique to collaborative perception. In single-agent tasks, quantization errors remain local. In contrast, collaborative perception suffers from the fact that quantization noise from $N$ heterogeneous agents compounds non-linearly during aggregation. Our proposed collaboration-aware calibration and sampling strategy is quite important to keep the quantization not in local level but in global level to ensure a global performance preservation. By forcing the calibration to cover the full dynamic range of agent density (from isolated single-agent cases to dense multi-agent scenarios), we ensure global system stability across dynamic graph structures of multi-agent collaborations. Thus, this is a domain-specific structural adaptation necessary to prevent fusion collapse in different domains.
> > >
> > > Taken together, we posit that our Multi-Agent-Aware Calibration Procedure and Alignment strategies represent a key methodological novelty necessitated by the unique challenges of the CP system. These are not generic applications, but tailored solutions that effectively resolve the domain-specific challenge of fusion-layer error amplification with diverse heterogeneous agents. We appreciate the reviewer for recognizing this work as the first fully quantized V2X system; we believe this framework not only solves immediate deployment hurdles but establishes a vital foundation for future explorations.

---

> > > ### Author Response · Authors · 2025-11-24
> > > **Response to Generalization to Unseen sensors**
> > >
> > > We acknowledge the reviewer's point that generalizing to unseen sensors is a valuable long-term goal. However, we emphasize that QuantV2X is the first work to attempt full quantization of a collaborative system. Even with known sensor-encoder pairs, the technical challenge is immense given both the model-level and communication-level efficiency requirements. Specifically, the non-linear amplification of quantization noise during the fusion of heterogeneous agents creates a critical stability bottleneck that exists even when sensor configurations are known. We respectfully argue that solving this stability problem for known fleets is the necessary prerequisite for the future generalization to unseen sensors; if the system collapses under quantization for known pairs, it will undoubtedly fail for unknown ones. Therefore, we deliberately isolated the "quantization stability" challenge from the "unseen generalization" challenge to establish a robust baseline.
> > > In the meanwhile, we will add the discussion of this future work in the revised version in the appendix section. Specifically, we will outline how the QuantV2X foundation can be integrated with advanced open heterogeneity frameworks, such as HEAL or STAMP, to explicitly examine their unseen agent generalization capabilities within a fully quantized, resource-constrained environment. We believe that successfully establishing the first working fully quantized system validates the specific contribution of this paper.

---

> ### Author Response · Authors · 2025-11-20
> **Response to Minor Questions**
>
> ### 1.Why were KL Divergence and L2 loss chosen for heterogeneity and spatial alignment losses, respectively?
>
> KL Divergence (Heterogeneity): The feature maps are high-dimensional distributions representing semantic content. KL Divergence is a standard and effective metric for matching such distributions, ensuring the semantic information from heterogeneous encoders is aligned.
> L2 Loss (Spatial): The bounding box representations are continuous spatial coordinates (e.g., x, y, z, w, l, h). L2 (Euclidean) distance is the most direct and effective loss for minimizing the error in these spatial-geometric predictions, ensuring precise localization.
>
> ### 2. For the spatial alignment loss, the paper mentions bounding box representations, but the code uses only the regression head output of the multi-class detection head—why?
>
> Our spatial alignment loss specifically targets the regression head output (which predicts bounding box parameters) and not the classification head output. This is an intentional design choice to decouple the alignment: we use L2 loss for the spatial/geometric regression task, while the semantic/classification information is aligned via the KL divergence loss on the feature maps.
>
> ### 3. Why is a uniform distribution used to model communication latency?
>
> Such modeling is adopted as a common choice where we follow the experimental protocol of prior works V2X-ViT and V2XPnP, which models this communication uncertainty as a uniform distribution. Specific details could be referred to in the Page 22 of V2X-ViT paper (https://arxiv.org/pdf/2203.10638).
>
> ### 4. What are the training times for Stage 2 and Stage 3?
> Stage 2 training takes 20 epochs and stage 3 training takes 10 epochs (more details in Appendix E.3).

---

> ### Author Response · Authors · 2025-11-20
> **Additional experiments on decomposition of system-level latency components**
>
> Thanks for your valuable advice. We provide the following experiments on decomposition of latency components to analyze the effect of communication-level and model-level inefficiencies on the system-level performance. We report the inference time and their corresponding system-level performance as below.
>
> ### Table A: System-Level Latency Decomposition (ms)
> *Note: $T_{comp} = T_{loc} + T_{fusion}$ (Computation Latency)*
>
> | Method  | Pyramid ($T_{comm}$ / $T_{comp}$) | F-Cooper ($T_{comm}$ / $T_{comp}$) | AttFuse ($T_{comm}$ / $T_{comp}$) |
> | :--- | :--- | :--- | :--- |
> | **Full-Precision (FP)** | 286.8 / 59.5 | 511.2 / 47.4 | 511.2 / 53.3 |
> | **FP + x16 Comp.** | 68.2 / 59.5 | 121.7 / 47.4 | 121.7 / 53.3 |
> | **Codebook Only** | 12.9 / 59.5 | 13.6 / 47.4 | 13.6 / 53.3 |
> | **Quantization Only** | 87.0 / **27.1** | 158.9 / **20.2** | 158.9 / **21.6** |
> | **QuantV2X (Full)** | **12.9** / **27.1** | **13.6** / **20.2** | **13.6** / **21.6** |
>
> ### Table B: System-Level Accuracy (mAP@30 / mAP@50)
> *Note: "Ideal Accuracy" refers to ideal performance with zero latency and no feature compression.*
>
> | Config | Pyramid Fusion | F-Cooper | AttFuse |
> | :--- | :--- | :--- | :--- |
> | **Ideal Accuracy** | 53.8 / 43.5 | 47.3 / 37.9 | 48.3 / 38.5 |
> | **Full-Precision (FP)** | 43.1 / 34.8 | 40.3 / 31.8 | 43.4 / 33.9 |
> | **FP + x16 Comp.** | 48.8 / 38.0 | 41.4 / 32.2 | 44.6 / 35.6 |
> | **Codebook Only** | 51.6 / 41.9 | 44.5 / 33.0 | 45.7 / 34.8 |
> | **Quantization Only** | 48.7 / 38.3 | 41.7 / 32.0 | 43.9 / 34.1 |
> | **QuantV2X (Full)** | **52.6 / 42.2** | **45.1 / 35.0** | **46.5 / 35.9** |
>
> **Key Observations and Insights**:
>
> **1. Insight on communication-level efficiency (impact of codebook)**: Comparing "Full-Precision" to "Codebook Only," we observe that drastically reducing $T_{comm}$ (e.g., 286ms $\rightarrow$ 12.9ms for Pyramid) yields the single largest increase in accuracy (+8.5 mAP@30). This confirms that data freshness is paramount in V2X. Furthermore, the codebook module is also more effective than the standard x16 compression method. For Pyramid Fusion, x16 Compression yields 48.8 mAP@30 with a 68.2ms transmission time. In contrast, our Codebook method yields 51.6 mAP@30 with only 12.9ms transmission time. This demonstrates that learning a discrete codebook is far more efficient for V2X feature transmission than standard compression algorithms.
>
> **2. Insight on model-level efficiency (impact of quantization):** While "Codebook Only" improves results significantly, it still hits a ceiling because the computation time ($T_{comp}$) remains high (~60ms). By adding "Quantization" (our full pipeline), we cut $T_{comp}$ by roughly 55% (59.5ms $\rightarrow$ 27.1ms). This combined reduction allows QuantV2X to achieve an mAP of 52.6, which is remarkably close to the theoretical "Ideal Accuracy" of 53.8, proving that both modules are essential for a fully optimized system.
>
> **3. Generalization across different fusion architectures**:
> Crucially, these observations are consistent across all three fusion methods (Pyramid, F-Cooper, AttFuse). For every architecture, the "Full QuantV2X" setting provides the highest accuracy and lowest total latency, validating that our framework is a generalizable solution for the domain, not specific to one model.
>
> Additional detail on experiment setting: since pyramid fusion is of BEV feature dimension (W, H, C) = (100, 356, 64) and other two fusion methods are of BEV feature dimension (W, H, C) = (64, 256, 256), the transmission speed is different for full-precision models on those fusion methods. Also, we set $n_L$ as 128 and $n_R$ as 1 for pyramid fusion based codebook and $n_L$ as 256 and $n_R$ as 2 for other two fusion methods based codebook. Thus the final codebook message size is slightly different which causes the latency difference.
>
> We will integrate these decomposed results and insights into the main paper to clearly illustrate how QuantV2X minimizes both transmission and computation overheads to maximize real-world performance.

---

> ### Author Response · Authors · 2025-11-20
> **Response to Minor Weaknesses**
>
> We sincerely appreciate the reviewer's carefulness in reviewing our paper. We will clarify the following points in our final revised version:
>
> **1. Insufficient explanation of how the combination weight $\alpha_n$ is generated (page 4).**: we will update the following explanation in our revised version: The combination weight $\alpha_r$ is generated by computing the distances between input feature segments and all codebook entries, converting these distances to logits, and applying Gumbel-Softmax to produce soft, differentiable weights during training (which become hard one-hot selections during inference) that are then used to reconstruct the quantized feature as a weighted combination of the selected codes from each codebook group.
>
> **2. The specific values of $n_L$ and $n_R$ used in QuantV2X are not stated.**: We use $n_L$ as 128 and $n_R$ as 1.
>
> **3. Poor visual distinction between solid and dashed lines in Figure 4’s legend.**: We will update the visual appearance in our final version.
>
> **4. Missing explanation for the “Bits (W/A)” notation in Figure 4.**: we set all Bits (W/A) to INT4/8 in Sec. 3.1. We will add this notation in our final version.
>
> **5. The distribution of pose errors is not described.**: we follow the standard evaluation protocol established in prior V2X literature (e.g., OPV2V, V2X-ViT, HEAL) and use  sample noises from Gaussian distribution added to the ground truth pose of each collaborating agent (positional or heading error). We will clarify this in our revised version.
>
> **6. Lack of explanation for the model used as the upper bound.**: we use pyramid fusion as the model for upper bound (ideal accuracy without considerations of system-level latency and feature compression).
>
> **7. Missing details on BEV feature dimensions (e.g., ego detection range, voxel size).**: for pyramid fusion model, the feature dimension (W, H, C) is (100, 352, 64), while for other models (Attfuse, F-Cooper, etc.) the feature dimension (W, H, C) is (64, 256, 256). Range and voxel size configurations are set to the default settings in the corresponding datasets.
>
> **8. The description of V2X-ReaLO is absent from the main text.**: We will add the following description in the appendix Sec. E: V2X-ReaLO is an open-source, ROS-based framework and dataset designed to deploy and evaluate cooperative perception algorithms on real-world vehicles and smart infrastructure. Unlike static benchmarks, it facilitates the online execution of intermediate fusion pipelines, enabling the rigorous validation of deployment-critical metrics under dynamic physical constraints.
>
> **9. The paper reports mAP as the evaluation metric on the V2X-ReaL dataset, but it does not specify which object classes were included in the detection task.**: object classes are car, pedestrian and truck. More details could be found at page 8 of V2X-Real paper (https://arxiv.org/pdf/2403.16034).
>
> We hope these clarifications help better convey the significance and contributions of our work. We would be very grateful if you would consider these additional points in evaluating your assessment.
>
> Thank you again for your time and consideration.

---

> ### Author Response · Authors · 2025-11-24
> **Response to AdaRound Ablation Study**
>
> ### Response on AdaRound
> We respectfully clarify that both AdaRound and the Alignment Module are not redundant; they play distinct, synergistic roles in solving the quantization challenge in CP system. Our ablation studies (Tables 1 & 2 below) reveal two critical insights.
>
> **Table 1: Ablation Studies on Alignment Module Effect over AdaRound in terms of Precision (AP@0.7) and Longer Distances (short range: 0-30m/middle range: 30-50m/long range: 50m+) in DAIR-V2X Dataset ($L_P + C_R$ configuration)**
>
> | Bits (W/A) | Config | AP@0.3 (0-30) | AP@0.3 (30-50) | AP@0.3 (50+) | AP@0.5 (0-30) | AP@0.5 (30-50) | AP@0.5 (50+) | AP@0.7 (0-30) | AP@0.7 (30-50) | AP@0.7 (50+) | AP@0.3 (Total) | AP@0.5 (Total) | AP@0.7 (Total) |
> | :--- | :--- | :--- | :--- | :--- | :--- | :--- | :--- | :--- | :--- | :--- | :--- | :--- | :--- |
> | FP32 / 32 | Full Prec | 86.8 | 82.1 | 63.3 | 82.2 | 76.1 | 54.9 | 68.5 | 57.9 | 36.7 | 75.1 | 68.2 | 51.0 |
> | INT4 / 8 | MinMax | 86.4 | 80.6 | 59.4 | 79.0 | 68.0 | 46.4 | 46.1 | 33.3 | 13.5 | 73.2 | 61.5 | 24.0 |
> | INT4 / 8 | + AdaRound | 84.9 | 80.2 | 60.2 | 79.2 | 73.3 | 51.1 | 49.0 | 41.0 | 22.0 | 72.8 | 65.1 | 34.7 |
> | INT4 / 8 | + Alignment | **86.0** | **80.8** | **62.5** | **80.8** | **73.9** | **53.7** | **53.3** | **41.4** | **27.9** | **74.2** | **66.7** | **38.3** |
>
> **Key insight 1: Alignment Module Solves the "Precision & Range" Challenge (Where AdaRound Hits a Ceiling)**:
> While AdaRound stabilizes the weights, it hits a performance ceiling on strict metrics. As shown in Table 1, AdaRound alone struggles with high-precision localization (AP@0.7) and long-range detection (>50m). Those are the specific areas where "fusion collapse" blurs object details. The alignment module provides a massive boost: for AP@0.7 (50m+), it improves performance from 22.0 (AdaRound) to 27.9 (Ours), a ~27% relative improvement. Those results indicate that the alignment module is quite important in preserving the downstream detection performance in longer-range scenarios with higher requirements of precision.
>
> ** Table 2: Ablation Study - AdaRound vs. Naive Rounding (Nearest Neighbor) **
>
> | Bits (W/A) | Configuration | AP@0.3 | AP@0.5 |
> | :--- | :--- | :--- | :--- |
> | FP32 / 32 | Full Precision | 75.1 | 68.2 |
> | INT4 / 8 | MinMax Baseline | 73.2 | 61.5 |
> | INT4 / 8 | + Naive Rounding |70.0 | 58.6 |
> | **INT4 / 8** | **+ Alignment Module** | **70.3** | **59.3** |
>
> **Key insight 2: AdaRound is necessary to preserve CP system in a locally optimized state, but with alignment module the quantized system is able to reach the globally optimized state**: We actually notice that in the early stage of the project that naive rounding method (e.g., nearest neighbor) demonstrates that standard nearest-neighbor quantization degrades the performance. Since naive rounding optimizes error locally for each weight based solely on magnitude, this creates a systematic rounding bias (e.g., consistently rounding up) that ignores the layer's global output distribution. In multi-agent scenario, when naive rounding is applied, the systematic bias from every agent compounds during this summation. This makes the noise level of the fused feature map so high that the semantic and geometric feature of the scene is destroyed, which is consequently aligned with our observation in Table 2 that applying the Alignment Module on top of Naive Rounding failed to recover performance. AdaRound on the other hand adapts the rounding to minimize **layer-wise** reconstruction error. This effectively lowers the rounding error and preserving the basic structural integrity of the model weights and creates a more alignable feature space for multi-agent fusion. However, as more agents collaborate, the compounding errors from different agents make the AdaRound not to best optimized state as it largely focuses on single-agent's local optimization. That's why the alignment module is designed to calibrate the quantized model to a **globally optimized state that it does not need to perform perfect layer-wise optimization but just ensures that your key objectives (e.g., heterogeneous feature property and final spatial property, especially the scenarios that require high precision in long-range distance) are well-preserved (as Table 1 supported)**.
>
> We will add those key insights to our revised version.
>
> We hope these clarifications help better convey the significance and contributions of our work. We would be very grateful if you would consider these additional points in evaluating your assessment. Thank you again for your time and consideration.

---

### Official Review · Reviewer_zipq · 2025-11-01

**Soundness:** 3
**Presentation:** 3
**Contribution:** 3
**Rating:** 4
**Confidence:** 5

**Summary:**

This paper introduces QuantV2X, the first fully quantized multi-agent system for cooperative perception in autonomous driving. By jointly quantizing both the neural network models and the inter-agent communication messages, QuantV2X significantly reduces computational cost, memory usage, and communication bandwidth while maintaining accuracy comparable to full-precision systems. The framework integrates post-training quantization, codebook-based message compression, and an alignment module to address feature degradation and cross-agent inconsistencies. Extensive experiments on real-world V2X datasets demonstrate that QuantV2X achieves up to 3.2× lower system latency and +9.5 mAP improvement, highlighting its strong potential for scalable and efficient real-world deployment of collaborative autonomous driving systems.

**Strengths:**

The work is original in its integration of model-side and communication-side quantization into a unified, deployment-oriented framework that directly tackles real-world challenges, the limitation of communication bandwidth, which is essential for enabling real-world deployment.

The paper is clearly written with a smooth and logical flow, making it easy to follow.

**Weaknesses:**

1. The paper claims that QuantV2X outperforms linear quantization; however, linear quantization is a very basic baseline that is rarely used in modern deployments. To make the comparison more convincing, the authors are encouraged to include results against stronger quantization baselines, such as non-linear or piece-wise quantization methods as reviewed in [1].

2. While the experiments demonstrate that QuantV2X preserves performance under ideal conditions, it remains unclear how robust the system is to real-world imperfections such as GPS inaccuracy, pose noise, or communication latency. Evaluations under these non-ideal scenarios would strengthen the paper’s claim of deployability and highlight the practical resilience of the proposed method.

3. The writing quality could be improved for readability. Several figure and table elements (e.g., Figure 2, Table 2, and Table 8) contain fonts that are too small to read clearly, and there is no vertical spacing before Table 1, making the layout appear compressed. Improving the figure scaling and spacing will enhance the paper’s overall clarity and professional appearance.

[1] Gholami, A., Kim, S., Dong, Z., Yao, Z., Mahoney, M. W., & Keutzer, K. (2022). A survey of quantization methods for efficient neural network inference. In Low-power computer vision (pp. 291-326). Chapman and Hall/CRC.

**Questions:**

Can this tokenization method be extended to broader tasks such as segmentation, tracking, or collaborative end-to-end driving?

---

> ### Author Response · Authors · 2025-11-19
>
> We thank Reviewer zipq for the constructive feedback. We are encouraged that the reviewer found our QuantV2X “original and essential” for real-world deployment. We address the concerns as follows:
>
> ### Weakness 1: Baseline comparison is too weak; linear quantization is a basic baseline.
>
> **Response**: Thank you for raising this important point. We agree that linear quantization alone is insufficient as a strong baseline. In fact, we did evaluate QuantV2X against two widely recognized and stronger quantization baselines, PD-Quant and LiDAR-PTQ, in the Appendix (Sec. D.2, Table 8). Both methods represent recent post-training quantization strategies and are significantly more competitive than naive linear quantization. As shown in **Table 8** (we paste the results here for your convenience), QuantV2X substantially outperforms both PD-Quant and LiDAR-PTQ under identical 4-bit weight / 8-bit activation configurations:
>
> *Table: Comparison with SOTA Quantization Methods (4-bit Weight / 8-bit Activation) on DAIR-V2X dataset*
>
> | Methods | Bits (W/A) | AP@30 | AP@50 | Efficiency (GPU hrs) |
> | :--- | :--- | :--- | :--- | :--- |
> | PD-Quant | 4 / 8 | 65.5 | 56.1 | 0.37 |
> | LiDAR-PTQ | 4 / 8 | 73.8 | 65.7 | 0.93 |
> | **QuantV2X (Ours)** | **4 / 8** | **74.2** | **66.7** | **0.38** |
>
> The results indicate that our methods achieve less performance gap compared to the full-precision model while requiring much less calibration time compared to other strong quantization baselines. In the revised version we will move this result from Appendix to the main paper to enhance the persuasiveness of our method.
>
> ### Weakness 2: Unclear robustness under real-world imperfections (GPS inaccuracy, pose noise, comms latency).
>
> **Response**:
> We appreciate this valuable suggestion. Our method is indeed designed to handle model-side and communication-side quantization, but we conducted a lot of experiments under different noisy conditions, and shared our results and insights in the main section of our paper for robustness:
>
> **Pose & GPS Error**: We model GPS inaccuracy and pose noise as equivalent perturbations (randomized shifts in $x, y, z$ and yaw). As shown in *Figure 4*, QuantV2X is highly robust: even under extreme noise (up to 0.5 std dev) where objects are $>50m$ away, our alignment module successfully preserves critical features, preventing the performance collapse often seen in standard models.
>
> **Communication Latency**: We described our experimental settings under *section 3.2.1 and 3.2.2*, which already includes simulation for communication latency. It is calculated based on the feature size and transmission rate. Our results in *Table 3* demonstrate that QuantV2X achieves up to 3.2× lower system latency (127.7 to 40.0ms) and +9.5 mAP improvement when considering real-world system-level latency compared to full-precision baseline.
>
> The above results show that QuantV2X maintains performance stability under these conditions, demonstrating resilience to practical imperfections in V2X deployment scenarios. Furthermore, our method is more resilient under real-world system-level latency and significantly outperforming full-precision baseline, indicating that **QuantV2X could achieve substantial reductions in cumulative system latency and communication overhead while maintaining competitive perception performance relative to an ideal full-precision baselines**.
>
> ### Weakness 3: Writing quality, figure readability, and layout issues.
>
> **Response**:
> Thank you for pointing this out! We acknowledge that some figures and tables (e.g., Fig. 2, Table 2, Table 8) are smaller than optimal due to space constraints. In the revised version, we will:
> enlarge fonts and increase visual spacing,
> improve figure scaling for clarity,
> adjust layout to make key results easier to read.
> These changes will be reflected in our final version of the paper which will significantly improve readability and presentation quality.
>
> *Question: Can this quantization method be extended to broader tasks (segmentation, tracking, end-to-end driving)?*
>
> Response:
> Yes. The QuantV2X framework quantizes the intermediate BEV feature representation, which serves as a shared framework for various downstream tasks. Since the quantization and communication compression operate at the feature and representation levels, the same mechanism can be integrated with other segmentation networks, multi-object tracking pipelines, and end-to-end planners as long as the model outputs or intermediate features can be quantized and transmitted. We respectfully argue that we are the first to design such quantized system in cooperative driving tasks. In the revised version, we plan to further validate its effectiveness on other tasks in future work.
>
> We hope these clarifications help better convey the significance and contributions of our work. We would be very grateful if you would consider these additional points in evaluating your assessment.
>
> Thank you again for your time and consideration.

---

> > ### Comment · Reviewer_zipq · 2025-11-27
> >
> > Thanks for the reviewers' rebuttal. The reviewer has answered my question well. I would like to raise the score to 6.

---

> ### Author Response · Authors · 2025-11-25
> **Hope to hear from your feedback!**
>
> Dear reviewer zipq,
>
> Hope this finds you well! We are writing to gently follow up to ensure you had a chance to check our detailed response to your review. In particular, addressing your concern regarding the strength of our baselines, we have incorporated comparisons against state-of-the-art post-training quantization methods (PD-Quant and LiDAR-PTQ). Our results demonstrate that QuantV2X significantly outperforms these methods under identical 4-bit weight / 8-bit activation settings, confirming the effectiveness of our unified pipeline beyond basic linear quantization. We have also provided the requested clarifications regarding system robustness to real-world imperfections (GPS inaccuracy, communication latency) and the method's potential for broader tasks.
>
> We believe these additions strongly support the validity of our approach and hope they fully address your concerns. We remain available if you have any further questions before the discussion period closes.

---

### Author Response · Authors · 2025-12-03
**Rebuttal Summary**

Dear Reviewers, Area Chairs, Program Chairs, and Senior Area Chairs,

We sincerely thank all reviewers for their thoughtful comments and constructive feedback. In the revised manuscript, we have carefully and comprehensively addressed every concern. To help the Area Chair quickly grasp the overall evaluation of our paper and main progress achieved during the rebuttal, we summarize them as follows:

### **Reviewer Consensus and Overall Evaluation**
We are pleased that reviewers view our work positively, recognizing the significance, soundness, strong empirical performance and deployment readiness of the proposed QuantV2X, the **first fully quantized system for V2X cooperative perception with innovative system-level efficiency optimization**. Overall, the evaluation is positive: **6/6/6/4 with Reviewer zipq raised their score to 6 during the rebuttal period**.

---
### **Table of Reviewer Status**

| Reviewer ID | Score | Confidence | Note |
| :--- | :--- | :--- | :--- |
| **zipq** | **6** (was 4) | 5 | **Raised score** after authors referred SOTA baselines (PD-Quant, LiDAR-PTQ) and clarified robustness to real-world imperfections. The authors moved the comparison with SOTA baselines from Appendix to the main paper in the revised version.|
| **HTY5** | **6** | 3 | Acknowledged QuantV2X's deployment-oriented design as "refreshing and directly relevant to real-world autonomous driving", "technically comprehensive and well-justified", and provide evidence of "good deployment readiness". Suggested quantitative trade-off analysis, which authors provided. |
| **CVYj** | **6** | 3 | Recognized QuantV2X as "Innovative gap-filling design" system with "Effective technical framework" and "Strong engineering value". |
| **d6fb** | **4** | 4 | Acknowledged "the paper addresses an important limitation in current collaborative perception research" with "high significance and valuable motivation", but concerns methodological novelty. Authors provided such new analysis and new ablation studies and explanations showing our method is a "is a reasonable solution to a domain-specific problem of feature misalignment in multi-agent fusion" (as reviewer HTY5 pointed out). |

---
### **Summarized Rebuttal to Reviewer Concerns**
**System-Level Efficiency Analysis (Latency Decomposition)**: Added results with a rigorous decomposition of system latency into Communication ($T_{comm}$) and Computation ($T_{comp}$) components and present the corresponding system performance gain brought by each component with validations across three fusion methods (Pyramid Fusion, AttFuse, F-Cooper). Location: Section 3.2 (System-Level Efficiency Analysis) and Figure 6.

**Concerns with Methodological Novelty**: Deviation of methodological novelty occurs among reviewers. For positive reviews, reviewer **HTY5** explicitly states QuantV2X is "technically comprehensive and well-justified" and alignment module "is a reasonable solution to a domain-specific problem of feature misalignment in multi-agent fusion", and **CVYj** recognizes QuantV2X as "Innovative gap-filling design" system with "Effective technical framework". We address reviewer **d6fb**'s concern on the novelty of alignment module with more explanations of motivations of design and clarified that our method is not a redundant application of standard quantization methods and is crucial for bringing more enhancement in long-range detection with higher precision requirement. Location: Appendix D.5. **However, we respect the research taste of each independent reviewer and we won't forcefully argue that our methodological novelty justifies this work meeting the acceptance threshold.**

**Lack of SOTA comparisons**: we move the SOTA comparisons from Appendix D.2 (older version) to Section 3.1.2 in revised paper for clarity. Location: 3.1.2.

**Writing and figure readability**:  we enlarge fonts and increase visual spacing, improve figure scaling for clarity, and adjust layout to make key results easier to read in the revised version.

**Improve methodological explanation and experimental setting**: we add additional terminology explanation and add more explanations on the experimental settings (dataset/benchmark introduction and hyper-parameter details).

---
Again, we are very grateful for the reviewers' comments, which we believe further strengthen the robustness of the work and the clarity of manuscript. **As the first work to explore full quantization in cooperative perception, we acknowledge the inherent limitations of this pioneering step. However, we hope QuantV2X serves as a catalyst for the community, facilitating a necessary shift in focus. We advocate for reframing cooperative perception research around system-level efficiency, latency, and deployability, a perspective that is critical for transitioning V2X from theoretical prototypes to scalable, real-world deployment**.

We thank the Area Chair for the time and care devoted to overseeing the process and considering our submission.

---

### Meta-Review · Area_Chair_4qEh · 2026-01-07

**Summary:**

The paper QuantV2X received a mixed evaluation with a final score profile of 6, 6, 6, 4.

While reviewers acknowledged the system's practical value, there is a consensus that the work represents an engineering integration rather than a methodological breakthrough.

• Engineering Value and Practicality (Strengths): Reviewers zipq, HTY5, and cVYj praised the work for addressing real-world deployment bottlenecks in cooperative perception. They highlighted the "strong engineering value" of the fully quantized pipeline, which reduces latency by 3.2× while maintaining accuracy. The reviewers appreciated the shift from "accuracy-centric" to "system-level efficiency".

• Methodological Novelty (Major Weakness): Reviewer d6fb (Score: 4) strongly dissented, arguing that the paper lacks sufficient novelty for a top-tier conference. This reviewer noted that the core components—Codebook learning for communication and AdaRound for Post-Training Quantization (PTQ)—are direct applications of existing techniques. Even the "Alignment Module," claimed as a contribution, was characterized as a standard knowledge distillation technique (similar to "FitNets") rather than a collaboration-specific innovation. Reviewer HTY5 also conceded that the quantization techniques are based on existing literature.

• Experimental Concerns: Reviewer d6fb pointed out that the "heterogeneity alignment" was tested only on known sensor-encoder pairs, failing to prove robustness to unseen configurations. Additionally, cVYj noted the narrow experimental design, which excluded complex modalities like radar and real-world variables like signal occlusion.


Recommendation: Reject

Despite the successful rebuttal raising one score to a 6, we recommend rejecting this paper. A "marginal accept" (score 6) without strong advocacy often indicates a paper that is technically correct but lacks the significance or novelty required for a top-tier conference. The current work is fundamentally an engineering effort—assembling off-the-shelf tools (AdaRound, Codebooks, Distillation)—rather than a scientific advancement in representation learning.

**Reviewer Concerns:**

Novelty concern still remains even after the rebuttal period.

**Reviewer Scores:**

One reviewer raised their score. Others remains the same.

---

### Decision · Program_Chairs · 2026-01-26

Reject